# A Policy Gradient Algorithm for Learning to Learn in Multiagent Reinforcement Learning

## Abstract

A fundamental challenge in multiagent reinforcement learning is to learn beneficial behaviors in a shared environment with other agents that are also simultaneously learning. In particular, each agent perceives the environment as effectively non-stationary due to the changing policies of other agents. Moreover, each agent is itself constantly learning, leading to natural nonstationarity in the distribution of experiences encountered. In this paper, we propose a novel meta-multiagent policy gradient theorem that directly accommodates for the non-stationary policy dynamics inherent to these multiagent settings. This is achieved by modeling our gradient updates to directly consider both an agent's own non-stationary policy dynamics and the non-stationary policy dynamics of other agents interacting with it in the environment. We find that our theoretically grounded approach provides a general solution to the multiagent learning problem, which inherently combines key aspects of previous state of the art approaches on this topic. We test our method on several multiagent benchmarks and demonstrate a more efficient ability to adapt to new agents as they learn than previous related approaches across the spectrum of mixed incentive, competitive, and cooperative environments.

## 1 Introduction

Learning in multiagent settings is inherently more difficult than single-agent learning because an agent interacts both with the environment and other agents (Buşoniu et al., 2010). Specifically, the fundamental challenge in multiagent reinforcement learning (MARL) is the difficulty of learning optimal policies in the presence of other simultaneously learning agents because their changing behaviors jointly affect the environment's transition and reward function. This dependence on non-stationary policies renders the Markov property invalid from the perspective of each agent, requiring agents to adapt their behaviors with respect to potentially large, unpredictable, and endless changes in the policies of fellow agents (Papoudakis et al., 2019). In such environments, it is also critical that agents adapt to the changing behaviors of others in a very sample-efficient manner as it is likely that their policy could update again after a small number of interactions (Al-Shedivat et al., 2018). Therefore, effective agents should consider the learning of other agents and adapt quickly to non-stationary behaviors. Otherwise, undesirable outcomes may arise when an agent is constantly lagging in its ability to deal with the current policies of other agents.

In this paper, we propose a new framework based on meta-learning for addressing the inherent non-stationarity of MARL. Meta-learning (also referred to as learning to learn) was recently shown to be a promising methodology for fast adaptation in multiagent settings. The framework by Al-Shedivat et al. (2018), for example, introduces a meta-optimization scheme by which a meta-agent can adapt more efficiently to changes in a new opponent's policy after collecting only a handful of interactions. The key idea underlying their meta-optimization is to model the meta-agent's learning process so that its updated policy performs better than an evolving opponent. However, their work does not directly consider the opponent's learning process in the meta-optimization, treating the evolving opponent as an external factor and assuming the meta-agent cannot influence the opponent's future policy. As a result, their work fails to consider an important property of MARL: the opponent is also a learning agent changing its policy based on trajectories collected by interacting with the meta-agent. As such, the meta-agent has an opportunity to influence the opponent's future policy by changing the distribution of trajectories, and the meta-agent can take advantage of this opportunity to improve its performance during learning.

**Our contribution.** With this insight, we develop a new *meta-multiagent policy gradient theorem (Meta-MAPG)* that directly models the learning processes of all agents in the environment within a single objective function. We start by extending the meta-policy gradient theorem of Al-Shedivat et al. (2018) based on the multiagent stochastic policy gradient theorem (Wei et al., 2018) to derive a novel meta-policy gradient theorem. This is achieved by removing the unrealistic implicit assumption of Al-Shedivat et al. (2018) that the learning of other agents in the environment is not dependent on an agent's own behavior. Interestingly, performing our derivation with this more general set of assumptions *inherently* results in an additional term that was not present in previous work by Al-Shedivat et al. (2018). We observe that this added term is closely related to the process of shaping the learning dynamics of other agents in the framework of Foerster et al. (2018a). As such, our work can be seen as contributing a theoretically grounded framework that unifies the collective benefits of previous work by Al-Shedivat et al. (2018) and Foerster et al. (2018a). Meta-MAPG is evaluated on a diverse suite of multiagent domains, including the full spectrum of mixed incentive, competitive, and cooperative environments. Our experiments demonstrate that Meta-MAPG consistently results in superior adaption performance in the presence of novel evolving agents.

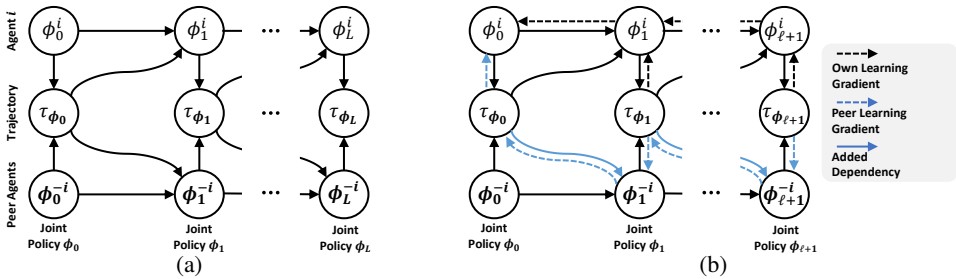

Figure 1: **(a)** A Markov chain of joint policies representing the inherent non-stationarity of MARL. Each agent updates its policy leveraging a Markovian update function, resulting in a change to the joint policy. **(b)** A probabilistic graph for Meta-MAPG. Unlike Meta-PG, our approach actively influences the future policies of other agents as well through the peer learning gradient.

## 2 PRELIMINARIES

Interactions between multiple agents can be represented by stochastic games (Shapley, 1953). Specifically, an $n$-agent stochastic game is defined as a tuple $\mathcal{M}_n = \langle \mathcal{I}, \mathcal{S}, \mathcal{A}, \mathcal{P}, \mathcal{R}, \gamma \rangle$; $\mathcal{I} = \{1, ..., n\}$ is the set of $n$ agents, $\mathcal{S}$ is the set of states, $\mathcal{A} = \times_{i \in \mathcal{I}} \mathcal{A}^i$ is the set of action spaces, $\mathcal{P} : \mathcal{S} \times \mathcal{A} \mapsto \mathcal{S}$ is the state transition probability function, $\mathcal{R} = \times_{i \in \mathcal{I}} \mathcal{R}^i$ is the set of reward functions, and $\gamma \in [0, 1)$ is the discount factor. We typeset sets in bold for clarity. Each agent $i$ executes an action at each timestep $t$ according to its stochastic policy $a_t^i \sim \pi^i(a_t^i | s_t, \phi^i)$ parameterized by $\phi^i$, where $s_t \in \mathcal{S}$. A joint action $\boldsymbol{a_t} = \{a_t^i, \boldsymbol{a_t^{-i}}\}$ yields a transition from the current state $s_t$ to the next state $s_{t+1} \in \mathcal{S}$ with probability $\mathcal{P}(s_{t+1} | s_t, \boldsymbol{a_t})$, where the notation $-i$ indicates all other agents with the exception of agent $i$. Agent $i$ then obtains a reward according to its reward function $r_t^i = \mathcal{R}^i(s_t, \boldsymbol{a_t})$. At the end of an episode, the agents collect a trajectory $\tau_{\boldsymbol{\phi}}$ under the joint policy with parameters $\boldsymbol{\phi}$, where $\tau_{\boldsymbol{\phi}} := (s_0, \boldsymbol{a_0}, \boldsymbol{r_0}, ..., \boldsymbol{r_H})$, $\boldsymbol{\phi} = \{\phi^i, \boldsymbol{\phi^{-i}}\}$ represents the joint parameters of all policies, $\boldsymbol{r_t} = \{r_t^i, \boldsymbol{r_t^{-i}}\}$ is the joint reward, and $H$ is the horizon of the trajectory or episode.

### 2.1 A MARKOV CHAIN OF POLICIES

The perceived non-stationarity in multiagent settings results from a distribution of sequential joint policies, which can be represented by a Markov chain (Al-Shedivat et al., 2018). Formally, a Markov chain of policies begins from a stochastic game between agents with an initial set of joint policies parameterized by $\boldsymbol{\phi_0}$. We assume that each agent updates its policy leveraging a Markovian update function that changes the policy after every $K$ trajectories. After this time period, each agent $i$ adapts its policy to maximize the expected return expressed as its value function:

$$V_{\boldsymbol{\phi_0}}^i(s_0) = \mathbb{E}_{\tau_{\boldsymbol{\phi_0}} \sim p(\tau_{\boldsymbol{\phi_0}} | \phi_0^i, \boldsymbol{\phi_0^{-i}})} \Big[ \sum_{t=0}^{H} \gamma^t r_t^i | s_0 \Big] = \mathbb{E}_{\tau_{\boldsymbol{\phi_0}} \sim p(\tau_{\boldsymbol{\phi_0}} | \phi_0^i, \boldsymbol{\phi_0^{-i}})} \Big[ G^i(\tau_{\boldsymbol{\phi_0}}) \Big], \qquad (1)$$

where $G^i$ denotes agent $i$'s discounted return from the beginning of an episode with initial state $s_0$. The joint policy update results in a transition from $\phi_0$ to the updated set of joint parameters $\phi_1$. The Markov chain continues for a maximum chain length of $L$ (see Figure 1a). This Markov chain perspective highlights the following inherent aspects of the experienced non-stationarity:

**Sequential dependency.** The future joint policy parameters $\phi_{1:L} = \{\phi_1, \ldots, \phi_L\}$ sequentially depend on $\phi_0$ since a change in $\tau_{\phi_0}$ results in a change in $\phi_1$, which in turn affects $\tau_{\phi_1}$ and all successive joint policy updates up to $\phi_L$.

**Controllable levels of non-stationarity.** As in Al-Shedivat et al. (2018) and Foerster et al. (2018a), we assume stationary policies during the collection of $K$ trajectories, and that the joint policy update happens afterward. In such a setting, it is possible to control the non-stationarity by adjusting the $K$ and $H$ hyperparameters: smaller $K$ and $H$ increase the rate that agents change their policies, leading to a higher degree of non-stationarity in the environment. In the limit of $K = H = 1$, all agents change their policy every step.

## 3 LEARNING TO LEARN IN MULTIAGENT REINFORCEMENT LEARNING

This section explores learning policies that can adapt quickly to non-stationarity in the policies of other agents in the environment. To achieve this, we leverage meta-learning and devise a new *meta-multiagent policy gradient theorem* that exploits the inherent sequential dependencies of MARL discussed in the previous section. Specifically, our meta-agent addresses this non-stationarity by considering its current policy's impact on its own adapted policies while actively influencing the future policies of other agents as well by inducing changes to the distribution of trajectories. In this section, we first outline the meta-optimization process in MARL and then discuss how the meta-policy gradient theorem of Al-Shedivat et al. (2018) optimizes for this objective while ignoring the dependence of the future policy of other agents on our current policy. Finally, we derive a new extension of this policy gradient theorem that explicitly leverages this dependence and discuss how to interpret the impact of the resulting form of the gradient.

### 3.1 GRADIENT BASED META-OPTIMIZATION IN MULTIAGENT REINFORCEMENT LEARNING

We formalize the meta-objective of MARL as optimizing meta-agent $i$'s initial policy parameters $\phi_0^i$ so that it maximizes the expected adaptation performance over a Markov chain of policies drawn from a stationary initial distribution of policies for the other agents $p(\phi_0^{-i})$:

$$\max_{\phi_0^i} \mathbb{E}_{p(\phi_0^{-i})} \left[ \sum_{\ell=0}^{L-1} V_{\phi_{0:\ell+1}}^i(s_0, \phi_0^i) \right], \tag{2}$$

$$\text{s.t.} \quad V_{\phi_{0:\ell+1}}^i(s_0, \phi_0^i) = \mathbb{E}_{\tau_{\phi_{0:\ell}} \sim p(\tau_{\phi_{0:\ell}} | \phi_{0:\ell}^i, \phi_{0:\ell}^{-i})} \left[ \mathbb{E}_{\tau_{\phi_{\ell+1}} \sim p(\tau_{\phi_{\ell+1}} | \phi_{\ell+1}^i, \phi_{\ell+1}^{-i})} \left[ G^i(\tau_{\phi_{\ell+1}}) \right] \right] \tag{3}$$

where $\tau_{\phi_{0:\ell}} = \{\tau_{\phi_0}, \ldots, \tau_{\phi_\ell}\}$, and $V_{\phi_{0:\ell+1}}^i(s_0, \phi_0^i)$ denotes the meta-value function. This meta-value function generalizes the notion of each agent's primitive value function for the current set of policies $V_{\phi_0}^i(s_0)$ over the length of the Markov chain of policies. In this work, as in Al-Shedivat et al. (2018), we follow the MAML (Finn et al., 2017) meta-learning framework. As such, we assume that the Markov chain of policies is governed by a policy gradient update function that corresponds to what is generally referred to as the inner-loop optimization in the meta-learning literature:

$$\begin{aligned} \phi_{\ell+1}^i &:= \phi_\ell^i + \alpha^i \nabla_{\phi_\ell^i} \mathbb{E}_{\tau_{\phi_\ell} \sim p(\tau_{\phi_\ell} | \phi_\ell^i, \phi_\ell^{-i})} \left[ G^i(\tau_{\phi_\ell}) \right], \\ \phi_{\ell+1}^{-i} &:= \phi_\ell^{-i} + \alpha^{-i} \nabla_{\phi_\ell^{-i}} \mathbb{E}_{\tau_{\phi_\ell} \sim p(\tau_{\phi_\ell} | \phi_\ell^i, \phi_\ell^{-i})} \left[ G^{-i}(\tau_{\phi_\ell}) \right], \end{aligned} \tag{4}$$

where $\alpha^i$ and $\alpha^{-i}$ denote the learning rates used by each agent in the environment.

### 3.2 THE META-POLICY GRADIENT THEOREM

Intuitively, if we optimize the meta-value function, we are searching for initial parameters $\phi_0^i$ such that successive inner-loop optimization steps with Equation (4) results in adapted parameters $\phi_{\ell+1}^i$ that can perform better than the updated policies of other agents with policy parameters $\phi_{\ell+1}^{-i}$ (see Figure 1b).

**Algorithm 1** Meta-Learning at Training Time

**Require:** $p(\phi_0^{-i})$: Distribution over other agents' initial policies; $\alpha, \beta$: Learning rates
1: Randomly initialize $\phi_0^i$
2: **while** $\phi_0^i$ has not converged **do**
3:     Sample a meta-train batch of $\phi_0^{-i} \sim p(\phi_0^{-i})$
4:     **for** each $\phi_0^{-i}$ **do**
5:         **for** $\ell = 0, \ldots, L$ **do**
6:             Sample and store trajectory $\tau_{\phi_\ell}$
7:             Compute $\phi_{\ell+1} = f(\phi_\ell, \tau_{\phi_\ell}, \alpha)$ from inner-loop optimization (Equation (4))
8:         **end for**
9:     **end for**
10:    Update $\phi_0^i \leftarrow \phi_0^i + \beta \sum_{\ell=0}^{L-1} \nabla_{\phi_0^i} V_{\phi_{0:\ell+1}}^i$ $(s_0, \phi_0^i)$ based on Equation (6)
11: **end while**

**Algorithm 2** Meta-Learning at Execution Time

**Require:** $p(\phi_0^{-i})$: Distribution over other agents' initial policies; $\alpha$: Learning rate; Optimized $\phi_0^{i*}$
1: Initialize $\phi_0^i \leftarrow \phi_0^{i*}$
2: Sample a meta-test batch of $\phi_0^{-i} \sim p(\phi_0^{-i})$
3: **for** each $\phi_0^{-i}$ **do**
4:     **for** $\ell = 0, \ldots, L$ **do**
5:         Sample trajectory $\tau_{\phi_\ell}$
6:         Compute $\phi_{\ell+1} = f(\phi_\ell, \tau_{\phi_\ell}, \alpha)$ from inner-loop optimization (Equation (4))
7:     **end for**
8: **end for**

In Deep RL, a very practical way to optimize a value function is by following its gradient. The work of Al-Shedivat et al. (2018) derived the meta-policy gradient theorem (Meta-PG) for optimizing a setup like this. However, it is important to note that they derived this gradient while making the implicit assumption to ignore the dependence of the future parameters of other agents on $\phi_0^i$:

$$\nabla_{\phi_0^i} V_{\phi_{0:\ell+1}}^i(s_0, \phi_0^i) = \mathbb{E}_{\tau_{\phi_{0:\ell}} \sim p(\tau_{\phi_{0:\ell}}|\phi_{0:\ell}^i, \phi_{0:\ell}^{-i})} \left[ \mathbb{E}_{\tau_{\phi_{\ell+1}} \sim p(\tau_{\phi_{\ell+1}}|\phi_{\ell+1}^i, \phi_{\ell+1}^{-i})} \right.$$
$$\left. \left[ \left( \underbrace{\nabla_{\phi_0^i} \log \pi(\tau_{\phi_0}|\phi_0^i)}_{\text{Current Policy}} + \underbrace{\sum_{\ell'=0}^{\ell} \nabla_{\phi_0^i} \log \pi(\tau_{\phi_{\ell'+1}}|\phi_{\ell'+1}^i)}_{\text{Own Learning}} \right) G^i(\tau_{\phi_{\ell+1}}) \right] \right]. \quad (5)$$

In particular, Meta-PG has two primary terms. The first term corresponds to the standard policy gradient with respect to the current policy parameters used during the initial trajectory. Meanwhile, the second term $\nabla_{\phi_0^i} \log \pi(\tau_{\phi_{\ell'+1}}|\phi_{\ell'+1}^i)$ explicitly differentiates through $\log \pi(\tau_{\phi_{\ell'+1}}|\phi_{\ell'+1}^i)$ with respect to $\phi_0^i$. This enables a meta-agent $i$ to model its own learning dynamics and account for the impact of $\phi_0^i$ on its eventual adapted parameters $\phi_{\ell'+1}^i$. As such, we can see how this term would be quite useful in improving adaptation across a Markov chain of policies. Indeed, it directly accounts for an agent's own learning process during meta-optimization in order to improve future performance.

### 3.3 THE META-MULTIAGENT POLICY GRADIENT THEOREM

In this section, we consider doing away with the implicit assumption from Al-Shedivat et al. (2018) discussed in the last section that we can ignore the dependence of the future parameters of other agents on $\phi_0^i$. Indeed, meta-agents need to account for both their own learning process and the learning processes of other peer agents in the environment to fully address the inherent non-stationarity of MARL. We will now demonstrate that our generalized gradient includes a new term explicitly accounting for the effect an agent's current policy has on the learned future policies of its peers.

**Theorem 1** (Meta-Multiagent Policy Gradient Theorem (Meta-MAPG)). *For any stochastic game* $\mathcal{M}_n$, *the gradient of the meta-objective function for agent $i$ at state $s_0$ with respect to the current parameters $\phi_0^i$ of stochastic policy $\pi$ evolving in the environment along with other peer agents using initial parameters $\phi_0^{-i}$ is:*

$$\nabla_{\phi_0^i} V_{\phi_{0:\ell+1}}^i(s_0, \phi_0^i) = \mathbb{E}_{\tau_{\phi_{0:\ell}} \sim p(\tau_{\phi_{0:\ell}}|\phi_{0:\ell}^i, \phi_{0:\ell}^{-i})} \left[ \mathbb{E}_{\tau_{\phi_{\ell+1}} \sim p(\tau_{\phi_{\ell+1}}|\phi_{\ell+1}^i, \phi_{\ell+1}^{-i})} \right[ \quad (6)$$
$$\left( \underbrace{\nabla_{\phi_0^i} \log \pi(\tau_{\phi_0}|\phi_0^i)}_{\text{Current Policy}} + \underbrace{\sum_{\ell'=0}^{\ell} \nabla_{\phi_0^i} \log \pi(\tau_{\phi_{\ell'+1}}|\phi_{\ell'+1}^i)}_{\text{Own Learning}} + \underbrace{\sum_{\ell'=0}^{\ell} \nabla_{\phi_0^i} \log \pi(\tau_{\phi_{\ell'+1}}|\phi_{\ell'+1}^{-i})}_{\text{Peer Learning}} \right) G^i(\tau_{\phi_{\ell+1}}) \right] \right]$$

*Proof.* See Appendix A for a detailed proof of Theorem 1. $\qquad\square$

**Probabilistic model perspective.** Probabilistic models for Meta-PG and Meta-MAPG are depicted in Figure 1b. As shown by the own learning gradient direction, a meta-agent $i$ optimizes $\phi_0^i$ by accounting for the impact of $\phi_0^i$ on its updated parameters $\phi_{1:\ell+1}^i$ and adaptation performance $G^i(\tau_{\phi_{\ell+1}})$. However, Meta-PG considers the other agents as an external factor that cannot be influenced by the meta-agent, as indicated by the absence of the dependence between $\tau_{\phi_{0:\ell}}$ and $\phi_{1:\ell+1}^{-i}$ in Figure 1b. As a result, the meta-agent loses an opportunity to influence the future policies of other agents and further improve its adaptation performance. By contrast, the peer learning term in Theorem 1 aims to additionally compute gradients through the sequential dependency between the agent's initial policy $\phi_0^i$ and the future policies of other agents in the environment $\phi_{1:\ell+1}^{-i}$ so that it can learn to change $\tau_{\phi_0}$ in a way that maximizes performance over the Markov chain of policies.

Interestingly, the peer learning term that naturally arises when taking the gradient in Meta-MAPG has been previously considered in the literature by Foerster et al. (2018a). In the Learning with Opponent Learning Awareness (LOLA) approach (Foerster et al., 2018a), this term was derived in an alternate way following a first order Taylor approximation with respect to the value function. Indeed, it is quite surprising to see how taking a principled policy gradient while leveraging a more general set of assumptions leads to a unification of the benefits of past works (Al-Shedivat et al., 2018; Foerster et al., 2018a) on adjusting to the learning behavior of other agents in MARL.

**Algorithm.** We provide pseudo-code for Meta-MAPG in Algorithm 1 for meta-training and Algorithm 2 for meta-testing. Note that Meta-MAPG is centralized during meta-training as it requires the policy parameters of other agents to compute the peer learning gradient. However, for settings where a meta-agent cannot access the policy parameters of other agents during meta-training, we provide a decentralized meta-training algorithm with opponent modeling, motivated by the approach used in Foerster et al. (2018a), in Appendix B that computes the peer learning gradient while leveraging only an approximation of the parameters of peer agents. Once meta-trained in either case, the adaptation to new agents during meta-testing is purely decentralized such that the meta-agent can decide how to shape other agents with its own observations and rewards alone.

# 4 RELATED WORK

The standard approach for addressing non-stationarity in MARL is to consider information about the other agents and reason about the effects of their joint actions (Hernandez-Leal et al., 2017). The literature on opponent modeling, for instance, infers opponents' behaviors and conditions an agent's policy on the inferred behaviors of others (He et al., 2016; Raileanu et al., 2018; Grover et al., 2018). Studies regarding the centralized training with decentralized execution framework (Lowe et al., 2017; Foerster et al., 2018b; Yang et al., 2018; Wen et al., 2019), which accounts for the behavior of others through a centralized critic, can also be classified into this category. While this body of work alleviates non-stationarity, it is generally assumed that each agent will have a stationary policy in the future. Because other agents can have different behaviors in the future as a result of learning (Foerster et al., 2018a), this incorrect assumption can cause sample inefficient and improper adaptation (see Example 1 in Appendix). In contrast, Meta-MAPG models the learning process of each agent in the environment, allowing a meta-learning agent to adapt efficiently.

Our approach is also related to prior work that considers the learning of other agents in the environment. This includes Zhang & Lesser (2010) who attempted to discover the best response adaptation to the anticipated future policy of other agents. Our work is also related, as discussed previously, to LOLA (Foerster et al., 2018a) and more recent improvements (Foerster et al., 2018c). Another relevant idea explored by Letcher et al. (2019) is to interpolate between the frameworks of Zhang & Lesser (2010) and Foerster et al. (2018a) in a way that guarantees convergence while influencing the opponent's future policy. However, all of these approaches only account for the learning processes of other agents and fail to consider an agent's own non-stationary policy dynamics as in the own learning gradient discussed in the previous section. Additionally, these papers do not leverage meta-learning. As a result, these approaches may require many samples to properly adapt to new agents.

Meta-learning (Schmidhuber, 1987; Bengio et al., 1992) has recently become very popular as a method for improving sample efficiency in the presence of changing tasks in the Deep RL literature (Wang et al., 2016a; Duan et al., 2016b; Finn et al., 2017; Mishra et al., 2017; Nichol & Schulman, 2018). See Vilalta & Drissi (2002); Hospedales et al. (2020) for in-depth surveys of meta-learning. In

Figure 2: Adaptation performance during meta-testing in mixed incentive ((**a**), (**b**)), competitive (**c**), and cooperative (**d**) environments. The results show that Meta-MAPG can successfully adapt to a new and learning peer agent throughout the Markov chain. Mean and 95% confidence interval computed for 10 random seeds for ((**a**), (**b**), (**c**)) and 5 random seeds for (**d**) are shown in figures.

particular, our work builds on the popular model agnostic meta-learning (MAML) framework (Finn et al., 2017) where gradient-based learning is used both for conducting so called inner-loop learning and to improve this learning by computing gradients through the computational graph. When we train our agents so that the inner loop can accommodate for a dynamic Markov chain of other agent policies, we are leveraging an approach that has recently become popular for supervised learning called meta-continual learning (Riemer et al., 2019; Javed & White, 2019; Spigler, 2019; Beaulieu et al., 2020; Caccia et al., 2020; Gupta et al., 2020). This means that our agent trains not just to adapt to a single set of policies during meta-training, but rather to adapt to a set of changing policies with Markovian updates. As a result, we avoid an issue of past work (Al-Shedivat et al., 2018) that required the use of importance sampling during meta-testing (see Appendix D.1 for more discussion).

## 5 EXPERIMENTS

We demonstrate the efficacy of Meta-MAPG on a diverse suite of multiagent domains, including the full spectrum of mixed incentive, competitive, and cooperative environments. To this end, we directly compare with the following baseline adaptation strategies:

1) Meta-PG (Al-Shedivat et al., 2018): A meta-learning approach that only considers how to improve its own learning. We detail our implementation of Meta-PG and a low-level difference with the implementation in the original paper by Al-Shedivat et al. (2018) in Appendix D.

2) LOLA-DiCE (Foerster et al., 2018c): An approach that only considers how to shape the learning dynamics of other agents in the environment through the Differentiable Monte-Carlo Estimator (DiCE) operation. Note that LOLA-DiCE is an extension of the original LOLA approach.

3) REINFORCE (Williams, 1992): A simple policy gradient approach that considers neither an agents own learning nor the learning processes of other agents. This baseline represents multiagent approaches that assume each agent leverages a stationary policy in the future.

In our experiments, we implement each method's policy leveraging an LSTM. The inner-loop updates are based on the policy gradient with a linear feature baseline (Duan et al., 2016a), and we use generalized advantage estimation (Schulman et al., 2016) with a learned value function for the meta-optimization. We also learn dynamic inner-loop learning rates during meta-training, as suggested in Al-Shedivat et al. (2018). We refer readers to Appendices C, D, E, H, and the source code in the supplementary material for the remaining details including selected hyperparameters.

**Question 1.** *Is it essential to consider both an agent's own learning and the learning of others?*

To address this question, we consider the classic iterated prisoner's dilemma (IPD) domain. In IPD, agents $i$ and $j$ act by either (C)ooperating or (D)efecting and receive rewards according to the mixed incentive payoff defined in Table 1. As in Foerster et al. (2018a), we model the state space as $s_0 = \varnothing$ and $s_t = a_{t-1}$ for $t \geq 1$.

|  | Agent $j$ | |
|---|---|---|
| Agent $i$ | $C$ | $D$ |
| $C$ | $(0.5, 0.5)$ | $(-1.5, 1.5)$ |
| $D$ | $(1.5, -1.5)$ | $(-0.5, -0.5)$ |

Table 1: IPD payoff table

For meta-learning, we construct a population of initial personas $p(\phi_0^{-i})$ that include cooperating personas (i.e., having a probability of cooperating between $0.5$ and $1.0$ at any state) and defecting

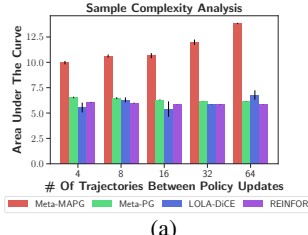 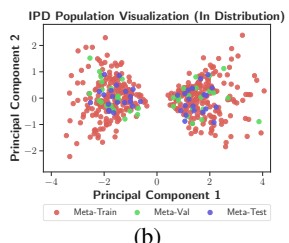 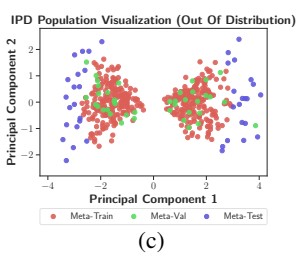

(a)  (b)  (c)

Figure 3: (**a**) Adaptation performance with a varying number of trajectories. Meta-MAPG achieves the best AUC in all cases and its performance generally improves with a larger $K$. Mean and 95% confidence interval are computed for 10 seeds. (**b**) and (**c**) Visualization of $j$'s initial policy for in distribution and out of distribution meta-testing, respectively, where the out of distribution split has a smaller overlap between the policies used for meta-training/validation and those used for meta-testing.

personas (i.e., having a probability of cooperating between 0 and 0.5 at any state). Figure 3b shows the population distribution utilized for training and evaluation. An agent $j$ is initialized randomly from the population and adapts its behavior leveraging the inner-loop learning process throughout the Markov chain (see Figure 6 in the appendix). Importantly, the initial persona of agent $j$ is hidden to $i$. Hence, an agent $i$ should: 1) adapt to a differently initialized agent $j$ with varying amounts of cooperation, and 2) continuously adapt with respect to the learning of $j$.

The adaptation performance during meta-testing when an agent $i$, meta-trained with either Meta-MAPG or the baseline methods, interacts with an initially cooperating or defecting agent $j$ is shown in Figure 2a and Figure 2b, respectively. In both cases, our meta-agent successfully infers the underlying persona of the other agent and adapts throughout the Markov chain obtaining higher rewards than our baselines. We observe that performance generally decreases as the number of joint policy update increases across all adaptation methods. This decrease in performance is expected as each model is playing with another agent that is also constantly learning. As a result, the other agent realizes it could potentially achieve more reward by defecting more often. Hence, to achieve good adaptation performance in IPD, an agent $i$ should attempt to shape $j$'s future policies toward staying cooperative as long as possible such that $i$ can take advantage, which is achieved by accounting for both an agent's own learning and the learning of other peer agents in Meta-MAPG.

We explore each adaptation method in more detail by visualizing the action probability dynamics throughout the Markov chain. In general, we observe that the baseline methods have converged to initially defecting strategies, attempting to get larger rewards than a peer agent $j$ in the first trajectory $\tau_{\phi_0}$. While this strategy can result in better initial performance than $j$, the peer agent will quickly change its policy so that it is defecting with high probability as well (see Figures 9 to 11 in the appendix). By contrast, our meta-agent learns to act cooperatively in $\tau_{\phi_0}$ and then take advantage by deceiving agent $j$ as it attempts to cooperate at future steps (see Figure 12 in the appendix).

**Question 2.** *How is adaptation performance affected by the number of trajectories between changes?*
We control the level of non-stationarity by adjusting the number of trajectories $K$ between updates (refer to Section 2.1). The results in Figure 3a shows that the area under the curve (AUC) (i.e., the reward summation during $\phi_{1:L}$) generally decreases when $K$ decreases in IPD. This result is expected since the inner-loop updates are based on the policy gradient, which can suffer from a high variance. Thus, with a smaller batch size, policy updates have a higher variance (leading to noisier policy updates). As a result, it is harder to anticipate and influence the future policies of other agents. Nevertheless, in all cases, Meta-MAPG achieves the best AUC.

**Question 3.** *Can Meta-MAPG generalize its learning outside the meta-training distribution?*
We have demonstrated that a meta-agent can generalize well and adapt to a new peer. However, we would like to investigate this further and see whether a meta-agent can still perform when the meta-testing distribution is drawn from a significantly different distribution in IPD. We thus evaluate Meta-MAPG and Meta-PG using both in distribution (as in the previous questions) and out of distribution personas for $j$'s initial policies (see Figures 3b and 3c). Meta-MAPG achieves an AUC of $13.77\pm0.25$ and $11.12\pm0.33$ for the in and out of distribution evaluation respectively. On the other hand, Meta-PG achieves an AUC of $6.13\pm0.05$ and $7.60\pm0.07$ for the in and out of distribution evaluation respectively. Variances are based on 5 seeds and we leveraged $K = 64$ for this experiment.

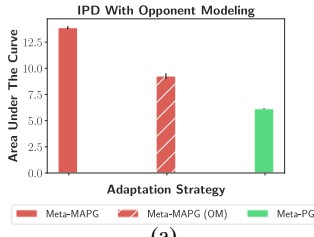 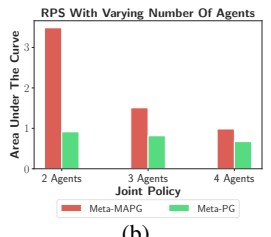 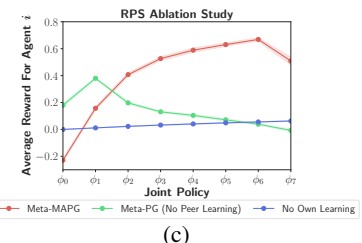

|(a)|(b)|(c)|

Figure 4: (**a**) Adaptation performance with opponent modeling (OM). Meta-MAPG with OM uses inferred policy parameters for peer agents, computing the peer learning gradient in a decentralized manner. (**b**) Adaptation performance with a varying number of agents in RPS. Meta-MAPG achieves the best AUC in all cases. (**c**) Ablation study for Meta-MAPG. Meta-MAPG achieves significantly better performance than ablated baselines with no own learning gradient and no peer learning gradient. The mean and $95\%$ confidence interval are computed using $5$ seeds in (**a**) and $10$ seeds in (**c**).

We note that Meta-MAPG's performance decreases during the out of distribution evaluation, but still consistently performs better than the baseline.

**Question 4.** *How does Meta-MAPG perform with decentralized meta-training?*
We compare the performance of Meta-MAPG with and without opponent modeling in Figure 4a. We note that Meta-MAPG with opponent modeling can infer policy parameters for peer agents and compute the peer learning gradient in a decentralized manner, performing better than the Meta-PG baseline. However, opponent modeling introduces noise in predicting the future policy parameters of peer agents because the parameters must be inferred by observing the actions they take alone without any supervision about the parameters themselves. Thus, as expected, meta-agents experience difficulty in correctly considering the learning process of peer agents, which leads to lower performance than Meta-MAPG with centralized meta-training.

**Question 5.** *How effective is Meta-MAPG in a fully competitive scenario?*
We have demonstrated the benefit of our approach in the mixed incentive scenario of IPD. Here, we consider another classic iterated game, rock-paper-scissors (RPS) with a fully competitive payoff table (see Table 2). In RPS, at each time step agents $i$ and $j$ can choose an action of either (R)ock, (P)aper, or (S)cissors. The state space is defined as $s_0 = \varnothing$ and $s_t = \boldsymbol{a_{t-1}}$ for $t \geq 1$.

|        |       | Agent $j$ |         |
|--------|-------|-----------|---------|
|        | $R$   | $P$       | $S$     |
| $R$    | $(0,0)$ | $(-1,1)$ | $(1,-1)$ |
| $P$    | $(1,-1)$ | $(0,0)$ | $(-1,1)$ |
| $S$    | $(-1,1)$ | $(1,-1)$ | $(0,0)$ |

Agent $i$

Table 2: RPS payoff table.

Similar to our meta-learning setup for IPD, we consider a population of initial personas $p(\boldsymbol{\phi_0^{-i}})$, including the rock persona (with a rock action probability between $1/3$ and $1.0$), the paper persona (with a paper action probability between $1/3$ and $1.0$), and the scissors persona (with a scissors action probability between $1/3$ and $1.0$). As in IPD, an agent $j$ is initialized randomly from the population and updates its policy based on the policy gradient with a linear baseline while interacting with $i$.

Figure 2c shows the adaptation performance during meta-testing. Similar to the IPD results, we observe that the baseline methods have effectively converged to win against the opponent $j$ in the first few trajectories. For instance, agent $i$ has a high rock probability when playing against $j$ with a high initial scissors probability (see Figures 13 to 15 in the appendix). This strategy, however, results in the opponent quickly changing its behavior toward the mixed Nash equilibrium strategy of $(1/3, 1/3, 1/3)$ for the rock, paper, and scissors probabilities. In contrast, our meta-agent learned to lose slightly in the first two trajectories $\tau_{\phi_{0:1}}$ to achieve much larger rewards in the later trajectories $\tau_{\phi_{2:7}}$ while relying on its ability to adapt more efficiently than its opponent (see Figure 16 in the appendix). Compared to the IPD results, we observe that it is more difficult for our meta-agent to shape $j$'s future policies in RPS possibly due to the fact that RPS has a fully competitive payoff structure, while IPD has a mixed incentive structure.

**Question 6.** *How effective is Meta-MAPG in settings with more than one peer?*
We note that the meta-multiagent policy gradient theorem is general and can be applied to scenarios with more than one peer. To validate this, we experiment with 3-player and 4-player RPS, where we consider sampling peers randomly from the entire persona population. Figure 4b shows a comparison against the Meta-PG baseline. We generally observe that the peer agents change their policies

toward the mixed Nash equilibrium more quickly as the number of agents increases, which results in decreased performance for all methods. Nevertheless, Meta-MAPG achieves the best performance in all cases and can clearly be easily extended to settings with a greater number of agents.

**Question 7.** *Is it necessary to consider both the own learning and peer learning gradient?*
Our meta-multiagent policy gradient theorem inherently includes both the own learning and peer learning gradient, but is it important to consider both terms? To answer this question, we conduct an ablation study and compare Meta-MAPG to two methods: one trained without the peer learning gradient and another trained without the own learning gradient. Note that not having the peer learning term is equivalent to Meta-PG, and not having the own learning term is similar to LOLA-DiCE but alternatively trained with a meta-optimization procedure. Figure 4c shows that a meta-agent trained without the peer learning term cannot properly exploit the peer agent's learning process. Also, a meta-agent trained without the own learning term cannot change its own policy effectively in response to anticipated learning by peer agents. By contrast, Meta-MAPG achieves superior performance by accounting for both its own learning process and the learning process of peer agents.

**Question 8.** *Does considering the peer learning gradient always improve performance?*
To answer this question, we experiment with a fully cooperative setting from the multiagent-MuJoCo benchmark (de Witt et al., 2020). Specifically, we consider the 2-Agent HalfCheetah domain, where the first and second agent control three joints of the back and front leg with continuous action spaces, respectively (see Figure 5). Both agents receive a joint reward corresponding to making the cheetah robot run to the right as soon as possible. Note that two agents are coupled *within* the cheetah robot, so accomplishing the objective requires close cooperation and coordination between them.

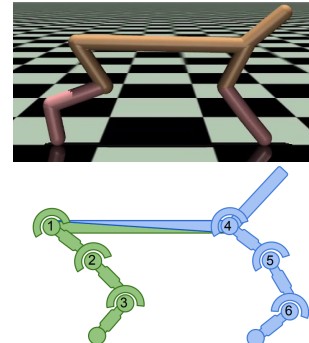

For meta-learning, we consider a population of teammates with varying degrees of expertise in running to the *left* direction. Specifically, we pre-train teammate $j$ and build a population based on checkpoints of its parameters during learning (see Figure 7 in Appendix). Then, during meta-learning, $j$ is randomly initialized from this population of policies. Importantly, the teammate must adapt its behavior in this setting because the agent has achieved the *opposite* skill compared to the true objective of moving to the right during pre-training. Hence, a meta-agent $i$ should succeed by both adapting to differently initialized teammates with varying expertise in moving the opposite direction, and guiding the teammate's learning process in order to coordinate eventual movement to the right.

Figure 5: 2-Agent HalfCheetah domain, where two agents are coupled within the robot and control the robot together. Graphic credit: de Witt et al. (2020).

Our results are displayed in Figure 2d. There are two notable observations. First, influencing peer learning does not help much in cooperative settings and Meta-MAPG performs similarly to Meta-PG. The peer learning gradient attempts to shape the future policies of other agents so that the meta-agent can take advantage. In IPD, for example, the meta-agent influenced $j$ to be cooperative in the future such that the meta-agent can act with a high probability of the defect action and receive higher returns. However, in cooperative settings, due to the joint reward, the teammate is already changing its policies in order to benefit the meta-agent, resulting in a less significant effect with respect to the peer learning gradient. Second, Meta-PG and Meta-MAPG outperform the other approaches of LOLA-DiCE and REINFORCE, achieving higher rewards when interacting with a new teammate.

## 6 CONCLUSION

In this paper, we have introduced Meta-MAPG which is a meta-learning algorithm that can adapt quickly to non-stationarity in the policies of other agents in a shared environment. The key idea underlying our proposed meta-optimization is to directly model both an agent's own learning process and the non-stationary policy dynamics of other agents. We evaluated our method on several multiagent benchmarks, including the full spectrum of mixed incentive, competitive, and cooperative environments. Our results indicate that Meta-MAPG is able to adapt more efficiently than previous state of the art approaches. We hope that our work can help provide the community with a theoretical foundation to build off for addressing the inherent non-stationarity of MARL in a principled manner. In the future, we plan to extend our approach to real-world scenarios, such as those including collaborative exploration between multiple agents (Chan et al., 2019).

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

# A  DERIVATION OF META-MULTIAGENT POLICY GRADIENT THEOREM

**Theorem 1** (Meta-Multiagent Policy Gradient Theorem (Meta-MAPG)). *For any stochastic game $\mathcal{M}_n$, the gradient of the meta-objective function for agent $i$ at state $s_0$ with respect to the current parameters $\phi_0^i$ of stochastic policy $\pi$ evolving in the environment along with the other peer agents using initial parameters $\phi_0^{-i}$ is:*

$$\nabla_{\phi_0^i} V_{\phi_{0:\ell+1}}^i(s_0, \phi_0^i) = \mathbb{E}_{\tau_{\phi_{0:\ell}} \sim p(\tau_{\phi_{0:\ell}}|\phi_{0:\ell}^i, \phi_{0:\ell}^{-i})} \Big[ \mathbb{E}_{\tau_{\phi_{\ell+1}} \sim p(\tau_{\phi_{\ell+1}}|\phi_{\ell+1}^i, \phi_{\ell+1}^{-i})} \Big[$$

$$\Big( \underbrace{\nabla_{\phi_0^i} \log \pi(\tau_{\phi_0}|\phi_0^i)}_{\text{Current Policy}} + \underbrace{\sum_{\ell'=0}^{\ell} \nabla_{\phi_0^i} \log \pi(\tau_{\phi_{\ell'+1}}|\phi_{\ell'+1}^i)}_{\text{Own Learning}} + \underbrace{\sum_{\ell'=0}^{\ell} \nabla_{\phi_0^i} \log \boldsymbol{\pi}(\tau_{\phi_{\ell'+1}}|\phi_{\ell'+1}^{-i})}_{\text{Peer Learning}} \Big) G^i(\tau_{\phi_{\ell+1}}) \Big] \Big]$$

*Proof.* We begin our derivation from the meta-value function defined in Equation (3). We expand the meta-value function with the state-action value and joint actions, assuming the conditional independence between agents' actions (Wen et al., 2019):

$$V_{\phi_{0:\ell+1}}^i(s_0, \phi_0^i) = \mathbb{E}_{\tau_{\phi_{0:\ell}} \sim p(\tau_{\phi_{0:\ell}}|\phi_{0:\ell}^i, \phi_{0:\ell}^{-i})} \Big[ \mathbb{E}_{\tau_{\phi_{\ell+1}} \sim p(\tau_{\phi_{\ell+1}}|\phi_{\ell+1}^i, \phi_{\ell+1}^{-i})} \big[ G^i(\tau_{\phi_{\ell+1}}) \big] \Big]$$

$$= \mathbb{E}_{\tau_{\phi_{0:\ell}} \sim p(\tau_{\phi_{0:\ell}}|\phi_{0:\ell}^i, \phi_{0:\ell}^{-i})} \Big[ V_{\phi_{\ell+1}}^i(s_0) \Big] \tag{7}$$

$$= \mathbb{E}_{\tau_{\phi_{0:\ell}} \sim p(\tau_{\phi_{0:\ell}}|\phi_{0:\ell}^i, \phi_{0:\ell}^{-i})} \Big[ \sum_{a_0^i} \pi(a_0^i|s_0, \phi_{\ell+1}^i) \sum_{\boldsymbol{a}_0^{-i}} \boldsymbol{\pi}(\boldsymbol{a}_0^{-i}|s_0, \phi_{\ell+1}^{-i}) Q_{\phi_{\ell+1}}^i(s_0, \boldsymbol{a}_0) \Big],$$

where $Q_{\phi_{\ell+1}}^i(s_0, \boldsymbol{a}_0)$ denotes the state-action value under the joint policy with parameters $\phi_{\ell+1}$ at state $s_0$ with joint action $\boldsymbol{a}_0$. In Equation (7), we note that both $\phi_{1:\ell}^i$ and $\phi_{1:\ell}^{-i}$ depend on $\phi_0^i$. Considering the joint update from $\phi_0$ to $\phi_1$, for simplicity, we can write the gradients in the inner-loop (Equation (4)) based on the multiagent stochastic policy gradient theorem (Wei et al., 2018):

$$\nabla_{\phi_0^{-i}} \mathbb{E}_{\tau_{\phi_0} \sim p(\tau_{\phi_0}|\phi_0^i, \phi_0^{-i})} \big[ G^i(\tau_{\phi_0}) \big] = \sum_s \rho_{\phi_0}(s) \sum_{a^i} \nabla_{\phi_0^i} \pi(a^i, \phi_0^i) \sum_{\boldsymbol{a}^{-i}} \boldsymbol{\pi}(\boldsymbol{a}^{-i}|s, \phi_0^{-i}) Q_{\phi_0}^i(s, \boldsymbol{a}),$$

$$\nabla_{\phi_0^{-i}} \mathbb{E}_{\tau_{\phi_0} \sim p(\tau_{\phi_0}|\phi_0^i, \phi_0^{-i})} \big[ \boldsymbol{G}^{-i}(\tau_{\phi_0}) \big] = \sum_s \rho_{\phi_0}(s) \sum_{\boldsymbol{a}^{-i}} \nabla_{\phi_0^{-i}} \boldsymbol{\pi}(\boldsymbol{a}^{-i}|s, \phi_0^{-i}) \sum_{a^i} \pi(a^i|s, \phi_0^i) \boldsymbol{Q}_{\phi_0}^{-i}(s, \boldsymbol{a}), \tag{8}$$

where $\rho_{\phi_0}$ denotes the stationary distribution under the joint policy with parameters $\phi_0$. Importantly, the inner-loop gradients for an agent $i$ and its peers are a function of $\phi_0^i$. Hence, the updated joint policy parameter $\phi_1$ depends on $\phi_0^i$. Following Equation (8), the successive inner-loop optimization until $\phi_{\ell+1}$ results in dependencies between $\phi_0^i$ and $\phi_{1:\ell+1}^i$ and between $\phi_0^i$ and $\phi_{1:\ell+1}^{-i}$ (see Figure 1b). Having identified which terms are dependent on $\phi_0^i$, we continue from Equation (7) and derive the gradient of the meta-value function with respect to $\phi_0^i$ by applying the product rule:

$$\nabla_{\phi_0^i} V_{\phi_{0:\ell+1}}^i(s_0, \phi_0^i)$$

$$= \nabla_{\phi_0^i} \Big[ \mathbb{E}_{\tau_{\phi_{0:\ell}} \sim p(\tau_{\phi_{0:\ell}}|\phi_{0:\ell}^i, \phi_{0:\ell}^{-i})} \Big[ \sum_{a_0^i} \pi(a_0^i|s_0, \phi_{\ell+1}^i) \sum_{\boldsymbol{a}_0^{-i}} \boldsymbol{\pi}(\boldsymbol{a}_0^{-i}|s_0, \phi_{\ell+1}^{-i}) Q_{\phi_{\ell+1}}^i(s_0, \boldsymbol{a}_0) \Big] \Big]$$

$$= \nabla_{\phi_0^i} \Big[ \sum_{\tau_{\phi_{0:\ell}}} p(\tau_{\phi_{0:\ell}}|\phi_{0:\ell}^i, \phi_{0:\ell}^{-i}) \sum_{a_0^i} \pi(a_0^i|s_0, \phi_{\ell+1}^i) \sum_{\boldsymbol{a}_0^{-i}} \boldsymbol{\pi}(\boldsymbol{a}_0^{-i}|s_0, \phi_{\ell+1}^{-i}) Q_{\phi_{\ell+1}}^i(s_0, \boldsymbol{a}_0) \Big]$$

$$= \underbrace{\nabla_{\phi_0^i} \Big[ \sum_{\tau_{\phi_{0:\ell}}} p(\tau_{\phi_{0:\ell}}|\phi_{0:\ell}^i, \phi_{0:\ell}^{-i}) \Big] \sum_{a_0^i} \pi(a_0^i|s_0, \phi_{\ell+1}^i) \sum_{\boldsymbol{a}_0^{-i}} \boldsymbol{\pi}(\boldsymbol{a}_0^{-i}|s_0, \phi_{\ell+1}^{-i}) Q_{\phi_{\ell+1}}^i(s_0, \boldsymbol{a}_0)}_{\text{Term A}} +$$

$$\underbrace{\sum_{\tau_{\phi_{0:\ell}}} p(\tau_{\phi_{0:\ell}}|\phi_{0:\ell}^i, \phi_{0:\ell}^{-i}) \Big[ \sum_{a_0^i} \nabla_{\phi_0^i} \pi(a_0^i|s_0, \phi_{\ell+1}^i) \Big] \sum_{\boldsymbol{a}_0^{-i}} \boldsymbol{\pi}(\boldsymbol{a}_0^{-i}|s_0, \phi_{\ell+1}^{-i}) Q_{\phi_{\ell+1}}^i(s_0, \boldsymbol{a}_0)}_{\text{Term B}} +$$

$$\underbrace{\sum_{\tau_{\phi_{0:\ell}}} p(\tau_{\phi_{0:\ell}}|\phi_{0:\ell}^i, \phi_{0:\ell}^{-i}) \sum_{a_0^i} \pi(a_0^i|s_0, \phi_{\ell+1}^i) \Big[ \sum_{\boldsymbol{a}_0^{-i}} \nabla_{\phi_0^i} \boldsymbol{\pi}(\boldsymbol{a}_0^{-i}|s_0, \phi_{\ell+1}^{-i}) \Big] Q_{\phi_{\ell+1}}^i(s_0, \boldsymbol{a}_0)}_{\text{Term C}} +$$

$$\underbrace{\sum_{\tau_{\phi_{0:\ell}}} p(\tau_{\phi_{0:\ell}}|\phi_{0:\ell}^i, \phi_{0:\ell}^{-i}) \sum_{a_0^i} \pi(a_0^i|s_0, \phi_{\ell+1}^i) \sum_{a_0^{-i}} \pi(a_0^{-i}|s_0, \phi_{\ell+1}^{-i}) \left[ \nabla_{\phi_0^i} Q_{\phi_{\ell+1}}^i(s_0, a_0) \right]}_{\text{Term D}}. \quad (9)$$

We first focus on the derivative of the trajectories $\tau_{\phi_{0:\ell}}$ in Term A:

$$\nabla_{\phi_0^i} \left[ \sum_{\tau_{\phi_{0:\ell}}} p(\tau_{\phi_{0:\ell}}|\phi_{0:\ell}^i, \phi_{0:\ell}^{-i}) \right]$$

$$= \nabla_{\phi_0^i} \left[ \sum_{\tau_{\phi_0}} p(\tau_{\phi_0}|\phi_0^i, \phi_0^{-i}) \sum_{\tau_{\phi_1}} p(\tau_{\phi_1}|\phi_1^i, \phi_1^{-i}) \times \dots \times \sum_{\tau_{\phi_\ell}} p(\tau_{\phi_\ell}|\phi_\ell^i, \phi_\ell^{-i}) \right]$$

$$= \left[ \sum_{\tau_{\phi_0}} \nabla_{\phi_0^i} p(\tau_{\phi_0}|\phi_0^i, \phi_0^{-i}) \right] \prod_{\forall \ell' \in \{0,\dots,\ell\} \setminus \{0\}} \sum_{\tau_{\phi_{\ell'}}} p(\tau_{\phi_{\ell'}}|\phi_{\ell'}^i, \phi_{\ell'}^{-i}) + \quad (10)$$

$$\left[ \sum_{\tau_{\phi_1}} \nabla_{\phi_1^i} p(\tau_{\phi_1}|\phi_1^i, \phi_1^{-i}) \right] \prod_{\forall \ell' \in \{0,\dots,\ell\} \setminus \{1\}} \sum_{\tau_{\phi_{\ell'}}} p(\tau_{\phi_{\ell'}}|\phi_{\ell'}^i, \phi_{\ell'}^{-i}) + \dots +$$

$$\left[ \sum_{\tau_{\phi_\ell}} \nabla_{\phi_\ell^i} p(\tau_{\phi_\ell}|\phi_\ell^i, \phi_\ell^{-i}) \right] \prod_{\forall \ell' \in \{0,\dots,\ell\} \setminus \{\ell\}} \sum_{\tau_{\phi_{\ell'}}} p(\tau_{\phi_{\ell'}}|\phi_{\ell'}^i, \phi_{\ell'}^{-i}),$$

where the probability of collecting a trajectory under the joint policy with parameters $\phi_\ell$ is given by:

$$p(\tau_{\phi_\ell}|\phi_\ell^i, \phi_\ell^{-i}) = p(s_0) \prod_{t=0}^H \pi(a_t^i|s_t, \phi_\ell^i) \pi(a_t^{-i}|s_t, \phi_\ell^{-i}) \mathcal{P}(s_{t+1}|s_t, a_t). \quad (11)$$

Using Equation (11) and the log-derivative trick, Equation (10) can be further expressed as:

$$\left[ \mathbb{E}_{\tau_{\phi_0} \sim p(\tau_{\phi_0}|\phi_0^i, \phi_0^{-i})} \nabla_{\phi_0^i} \log \pi(\tau_{\phi_0}|\phi_0^i) \right] \prod_{\forall \ell' \in \{0,\dots,\ell\} \setminus \{0\}} \sum_{\tau_{\phi_{\ell'}}} p(\tau_{\phi_{\ell'}}|\phi_{\ell'}^i, \phi_{\ell'}^{-i}) +$$

$$\left[ \mathbb{E}_{\tau_{\phi_1} \sim p(\tau_{\phi_1}|\phi_1^i, \phi_1^{-i})} \nabla_{\phi_0^i} \left( \log \pi(\tau_{\phi_1}|\phi_1^i) + \log \pi(\tau_{\phi_1}|\phi_1^{-i}) \right) \right] \prod_{\forall \ell' \in \{0,\dots,\ell\} \setminus \{1\}} \sum_{\tau_{\phi_{\ell'}}} p(\tau_{\phi_{\ell'}}|\phi_{\ell'}^i, \phi_{\ell'}^{-i})$$

$$+ \dots + \quad (12)$$

$$\left[ \mathbb{E}_{\tau_{\phi_\ell} \sim p(\tau_{\phi_\ell}|\phi_\ell^i, \phi_\ell^{-i})} \nabla_{\phi_0^i} \left( \log \pi(\tau_{\phi_\ell}|\phi_\ell^i) + \log \pi(\tau_{\phi_\ell}|\phi_\ell^{-i}) \right) \right] \prod_{\forall \ell' \in \{0,\dots,\ell\} \setminus \{\ell\}} \sum_{\tau_{\phi_{\ell'}}} p(\tau_{\phi_{\ell'}}|\phi_{\ell'}^i, \phi_{\ell'}^{-i})$$

where the summations of the log-terms, such as $\nabla_{\phi_0^i} \left( \log \pi(\tau_{\phi_\ell}|\phi_\ell^i) + \log \pi(\tau_{\phi_\ell}|\phi_\ell^{-i}) \right)$ are *inherently* included due to the sequential dependencies between $\phi_0^i$ and $\phi_{1:\ell}$. We use the result of Equation (12) and organize terms to arrive at the following expression for Term A in Equation (9):

$$\mathbb{E}_{\tau_{\phi_{0:\ell}} \sim P(\tau_{\phi_{0:\ell}}|\phi_{0:\ell}^i, \phi_{0:\ell}^{-i})} \Big[$$

$$\left( \nabla_{\phi_0^i} \log \pi(\tau_{\phi_0}|\phi_0^i) + \sum_{\ell'=0}^{\ell-1} \nabla_{\phi_0^i} \log \pi(\tau_{\phi_{\ell'+1}}|\phi_{\ell'+1}^i) + \sum_{\ell'=0}^{\ell-1} \nabla_{\phi_0^i} \log \pi(\tau_{\phi_{\ell'+1}}|\phi_{\ell'+1}^{-i}) \right) \times$$

$$\sum_{a_0^i} \pi(a_0^i|s_0, \phi_{\ell+1}^i) \sum_{a_0^{-i}} \pi(a_0^{-i}|s_0, \phi_{\ell+1}^{-i}) Q_{\phi_{\ell+1}}^i(s_0, a_0) \Big]. \quad (13)$$

Coming back to Term B-D in Equation (9), repeatedly unrolling the derivative of the Q-function $\nabla_{\phi_0^i} Q_{\phi_{\ell+1}}^i(s_0, a_0)$ by following Sutton & Barto (1998) yields:

$$\mathbb{E}_{\tau_{\phi_{0:\ell}} \sim p(\tau_{\phi_{0:\ell}}|\phi_{0:\ell}^i, \phi_{0:\ell}^{-i})} \left[ \sum_s \rho_{\phi_{\ell+1}}(s) \sum_{a^i} \nabla_{\phi_0^i} \pi(a^i|s, \phi_{\ell+1}^i) \sum_{a^{-i}} \pi(a^{-i}|s, \phi_{\ell+1}^{-i}) Q_{\phi+1}^i(s, a) \right] + \quad (14)$$

$$\mathbb{E}_{\tau_{\phi_{0:\ell}} \sim p(\tau_{\phi_{0:\ell}}|\phi_{0:\ell}^i, \phi_{0:\ell}^{-i})} \left[ \sum_s \rho_{\phi_{\ell+1}}(s) \sum_{a^{-i}} \nabla_{\phi_0^i} \pi(a^{-i}|s, \phi_{\ell+1}^{-i}) \sum_{a^i} \pi(a^i|s, \phi_{\ell+1}^i) Q_{\phi_{\ell+1}}^i(s, a) \right],$$

which adds the consideration of future joint policy $\phi_{\ell+1}$ to Equation (13). Finally, we summarize Equations (13) and (14) together and express in expectations:

$$\nabla_{\phi_0^i} V_{\phi_{0:\ell+1}}^i(s_0, \phi_0^i) = \mathbb{E}_{\tau_{\phi_{0:\ell}} \sim p(\tau_{\phi_{0:\ell}}|\phi_{0:\ell}^i, \phi_{0:\ell}^{-i})} \Big[ \mathbb{E}_{\tau_{\phi_{\ell+1}} \sim p(\tau_{\phi_{\ell+1}}|\phi_{\ell+1}^i, \phi_{\ell+1}^{-i})} \Big[$$

$$\Big( \underbrace{\nabla_{\phi_0^i} \log \pi(\tau_{\phi_0}|\phi_0^i)}_{\text{Current Policy}} + \underbrace{\sum_{\ell'=0}^\ell \nabla_{\phi_0^i} \log \pi(\tau_{\phi_{\ell'+1}}|\phi_{\ell'+1}^i)}_{\text{Own Learning}} + \underbrace{\sum_{\ell'=0}^\ell \nabla_{\phi_0^i} \log \pi(\tau_{\phi_{\ell'+1}}|\phi_{\ell'+1}^{-i})}_{\text{Peer Learning}} \Big) G^i(\tau_{\phi_{\ell+1}}) \Big] \Big] \square$$

# B  META-MAPG WITH OPPONENT MODELING

---

**Algorithm 3** Meta-Learning at Training Time with Opponent Modeling

---

**Require:** $p(\phi_0^{-i})$: Distribution over other agents' initial policies; $\alpha, \beta, \hat{\alpha}$: Learning rates
1: Randomly initialize $\phi_0^i$
2: **while** $\phi_0^i$ has not converged **do**
3:     Sample a meta-train batch of $\phi_0^{-i} \sim p(\phi_0^{-i})$
4:     **for** each $\phi_0^{-i}$ **do**
5:         Randomly initialize $\hat{\phi}_0^{-i}$
6:         **for** $\ell = 0, ..., L$ **do**
7:             Sample and store trajectory $\tau_{\phi_\ell}$
8:             Approximate $\hat{\phi}_\ell^{-i} = f(\hat{\phi}_\ell^{-i}, \tau_{\phi_\ell}, \hat{\alpha})$ using opponent modeling (Algorithm 4)
9:             Compute $\phi_{\ell+1} = f(\phi_\ell, \tau_{\phi_\ell}, \alpha)$ from inner-loop optimization (Equation (4))
10:            Compute $\hat{\phi}_{\ell+1}^{-i} = f(\hat{\phi}_\ell^{-i}, \tau_{\phi_\ell}, \alpha)$ from inner-loop optimization (Equation (4))
11:        **end for**
12:    **end for**
13:    Update $\phi_0^i \leftarrow \phi_0^i + \beta \sum_{\ell=0}^{L-1} \nabla_{\phi_0^i} V_{\phi_{0:\ell+1}}^i(s_0, \phi_0^i)$ based on Equation (6) and $\hat{\phi}_{1:L}^{-i}$
14: **end while**

---

---

**Algorithm 4** Opponent Modeling

---

1: **procedure** OPPONENT MODELING($\hat{\phi}_\ell^{-i}, \tau_{\phi_\ell}, \hat{\alpha}$)
2:     **while** $\hat{\phi}_\ell^{-i}$ has not converged **do**
3:         Compute log-likelihood $\mathcal{L}_{\text{likelihood}} = f(\hat{\phi}_\ell^{-i}, \tau_{\phi_\ell})$ based on Equation (15)
4:         Update $\hat{\phi}_\ell^{-i} \leftarrow \hat{\phi}_\ell^{-i} + \hat{\alpha} \nabla_{\hat{\phi}_\ell^{-i}} \mathcal{L}_{\text{likelihood}}$
5:     **end while**
6:     **return** $\hat{\phi}_\ell^{-i}$
7: **end procedure**

---

In this section, we explain Meta-MAPG with opponent modeling for settings where a meta-agent cannot access the policy parameters of its peers during meta-training. Our decentralized meta-training method in Algorithm 3 replaces the other agents' true policy parameters $\phi_{1:L}^{-i}$ with inferred parameters $\hat{\phi}_{1:L}^{-i}$ in computing the peer learning gradient. Specifically, we follow Foerster et al. (2018a) for opponent modeling and estimate $\hat{\phi}_\ell^{-i}$ from $\tau_{\phi_\ell}$ using log-likelihood $\mathcal{L}_{\text{likelihood}}$ (Line 8 in Algorithm 3):

$$\mathcal{L}_{\text{likelihood}} = \sum_{t=0}^{H} \log \pi^{-i}(a_t^{-i} | s_t, \hat{\phi}_\ell^{-i}), \tag{15}$$

where $s_t, a_t^{-i} \in \tau_{\phi_\ell}$. A meta-agent can obtain $\hat{\phi}_{1:L}^{-i}$ by iteratively applying the opponent modeling procedure until the maximum chain length of $L$. We also apply the inner-loop update with the Differentiable Monte-Carlo Estimator (DiCE) (Foerster et al., 2018c) to the inferred policy parameters of peer agents (Line 10 in Algorithm 3). By applying DiCE, we can save the sequential dependencies between $\phi_0^i$ and updates to the policy parameters of peer agents $\hat{\phi}_{1:L}^{-i}$ in a computation graph and compute the peer learning gradient efficiently via automatic-differentiation (Line 13 in Algorithm 3).

# C  ADDITIONAL IMPLEMENTATION DETAILS

## C.1  NETWORK STRUCTURE

Our neural networks for the policy and value function consist of a fully-connected input layer with 64 units followed by a single-layer LSTM with 64 units and a fully-connected output layer. We reset the LSTM states to zeros at the beginning of trajectories and retain them until the end of episodes. The LSTM policy outputs a probability for the Bernoulli distribution in the iterated games (i.e., IPD,

RPS). For the 2-Agent HalfCheetah domain, the policy outputs a mean and variance for the Gaussian distribution. We empirically observe that no parameter sharing between the policy and value network results in more stable learning than sharing the network parameters.

## C.2 OPTIMIZATION

We detail additional important notes about our implementation:

- We apply the linear feature baseline (Duan et al., 2016a) and generalized advantage estimation (GAE) (Schulman et al., 2016) during the inner-loop and outer-loop optimization, respectively, to reduce the variance in the policy gradient.

- We use DiCE (Foerster et al., 2018c) to compute the peer learning gradient efficiently. Specifically, we apply DiCE during the inner-loop optimization and save the sequential dependencies between $\phi_0^i$ and $\phi_{1:L}^{-i}$ in a computation graph. Because the computation graph has the sequential dependencies, we can compute the peer learning gradient by the backpropagation of the meta-value function via the automatic-differentiation toolbox.

- Learning from diverse peers can potentially cause conflicting gradients and unstable learning. In IPD, for instance, a strategy to adapt against cooperating peers can be completely opposite to the adaptation strategy against defecting peers, resulting in conflicting gradients. To address this potential issue, we use the projecting conflicting gradients (PCGrad) (Yu et al., 2020) during the outer-loop optimization. We also have tested the baseline methods with PCGrad.

- We use a distributed training to speed up the meta-optimization. Each thread interacts with a Markov chain of policies until the chain horizon and then computes the meta-optimization gradients using Equation (6). Then, similar to Mnih et al. (2016), each thread asynchronously updates the shared meta-agent's policy and value network parameters.

# D  ADDITIONAL BASELINE DETAILS

We train all adaptation methods based on a meta-training set until convergence. We then measure the adaptation performance on a meta-testing set using the best-learned policy determined by a meta-validation set.

## D.1  META-PG

We have improved the Meta-PG baseline itself beyond its implementation in the original work (Al-Shedivat et al., 2018) to further isolate the importance of the peer learning gradient term. Specifically, compared to Al-Shedivat et al. (2018), we make the following theoretical contributions to build on:

**Underlying problem statement.**  Al-Shedivat et al. (2018) bases their problem formulation off that of multi-task / continual single-agent RL. In contrast, ours is based on a general stochastic game between $n$ agents (Shapley, 1953).

**A Markov chain of joint policies.**  Al-Shedivat et al. (2018) treats an evolving peer agent as an external factor, resulting in the absence of the sequential dependencies between a meta-agent's current policy and the peer agents' future policies in the Markov chain. However, our important insight is that the sequential dependencies exist in general multiagent settings as the peer agents are also learning agents based on trajectories by interacting with a meta-agent (see Figure 1b).

**Meta-objective.**  The meta-objective defined in Al-Shedivat et al. (2018) is based on single-agent settings. In contrast, our meta-objective is based on general multiagent settings (see Equations (2) to (4)).

**Meta-optimization gradient.**  Compared to Al-Shedivat et al. (2018), our meta-optimization gradient inherently includes the additional term of the peer learning gradient that considers how an agent can directly influence the learning process of other agents.

**Importance sampling.**  Compared to Al-Shedivat et al. (2018), we avoid using the importance sampling during meta-testing by modifying the meta-value function. Specifically, the framework uses a meta-value function on a pair consecutive joint policies, denoted $V_{\phi_{\ell:\ell+1}}^i(s_0, \phi_0^i)$, which assumes

initializing every $\phi_\ell^i$ from $\phi_0^i$. However, as noted in Al-Shedivat et al. (2018), this assumption requires interacting with the same peers multiple times and is often impossible during meta-testing. To address this issue, the framework uses the importance sampling correction during meta-testing. However, the correction generally suffers from high variance (Wang et al., 2016b). As such, we effectively avoid using the correction by initializing from $\phi_0^i$ only once at the beginning of Markov chains for both meta-training and meta-testing.

The above theoretical differences have resulted in an improved meta-agent that can learn to additionally affect future policies of other peer agents, achieving better results than the Meta-PG baseline in our experiments.

### D.2 LOLA-DiCE

We used an open-source PyTorch implementation for LOLA-DiCE.[1] We make minor changes to the code, such as adding the LSTM policy and value function.

## E ADDITIONAL EXPERIMENT DETAILS

### E.1 IPD

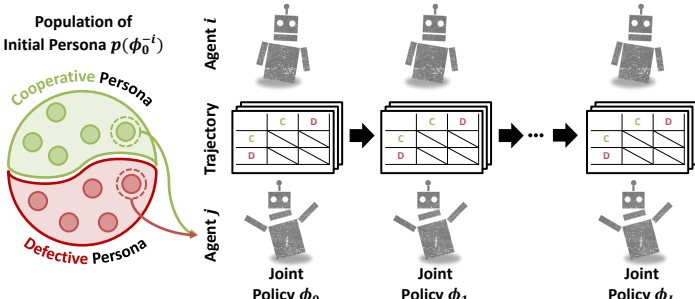

Figure 6: IPD meta-learning setup. An agent $j$'s policy is initialized randomly from the initial persona population $p(\phi_0^{-i})$ that includes various cooperating and defecting personas. The agent $j$ then updates its policy throughout the Markov chain, requiring an agent $i$ to adapt with respect to the learning of $j$.

We choose to represent the peer agent $j$'s policy as a tabular representation to effectively construct the population of initial personas $p(\phi_0^{-i})$ for the meta-learning setup. Specifically, the tabular policy has a dimension of $5$ that corresponds to the number of states in IPD. Then, we randomly sample a probability between $0.5$ and $1.0$ and a probability between $0$ and $0.5$ at each state to construct the cooperating and defecting population, respectively. As such, the tabular representation enables us to sample as many as personas but also controllable distribution $p(\phi_0^{-i})$ by merely adjusting the probability range. We sample a total of $480$ initial personas, including cooperating personas and defecting personas, and split them into $400$ for meta-training, $40$ for meta-validation, and $40$ for meta-testing. Figure 3b visualizes the distribution, where we used the principal component analysis (PCA) with two components.

### E.2 RPS

In RPS, we follow the same meta-learning setup as in IPD, except we sample a total of $720$ initial opponent personas, including rock, paper, and scissors personas, and split them into $600$ for meta-training, $60$ for meta-validation, and $60$ for meta-testing. Additionally, because RPS has three possible actions, we sample a rock preference probability between $1/3$ and $1.0$ for building the rock persona population, where the rock probability is larger than the other two action probabilities. We follow the same procedure for constructing the paper and scissors persona population.

---

[1]Available at `https://github.com/alexis-jacq/LOLA_DiCE`

### E.3 2-AGENT HALFCHEETAH

We used an open source implementation for multiagent-MuJoCo benchmark.[2] Agents in our experiments receive state observations that include information about all the joints. For the meta-learning setup, we pre-train a teammate $j$ with an LSTM policy that has varying expertise in moving to the left direction. Specifically, we train the teammate up to 500 train iterations and save a checkpoint at each iteration. Intuitively, as the number of train iteration increases, the teammate gains more expertise. We then use the checkpoints from 50 to 300 iterations as the meta-train/val and from 475 and 500 iterations as the meta-test distribution (see Figure 7). We construct the distribution with the gap to ensure that the meta-testing distribution has a sufficient difference to the meta-train/val so that we can test the generalization of our approach. Lastly, the teammate agent $j$
updates its policy based on the policy gradient with the linear feature baseline as in IPD and RPS.

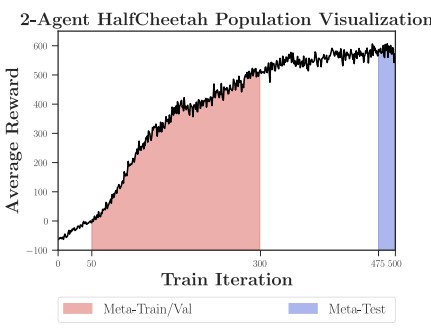

Figure 7: Visualization of a teammate $j$'s initial expertise in the 2-Agent HalfCheetah domain, where the meta-test distribution has a sufficient difference to meta-train/val.

## F IMPORTANCE OF PEER LEARNING

**Example 1.** *Failure to consider the learning process of the other agents can result in divergence of learning objectives.*

For example, consider a stateless zero-sum game playing between two agents. Agents $i$ and $j$ maximize simple value functions $V^i_\phi = \phi^i \phi^j$ and $V^j_\phi = -\phi^i \phi^j$ respectively, where $\phi^i, \phi^j \in \mathbb{R}$. In this game, there exists a unique Nash equilibrium at the origin (i.e., $\{\phi^i, \phi^j\} = \{0, 0\}$). We compare: 1) the standard approach that optimizes the value function in Equation (1) with the stationary assumption and 2) an approach that considers the learning process of others, such as the LOLA method. As Figure 8 shows, the standard approach diverges further from the equilibrium, resulting in worse results for both agents. The cause of the failure in this example is due to the stationary assumption that each agent assumes its opponent has the same behavior in the future (Letcher et al., 2019). In contrast, by considering the learning process of the opponent, the LOLA approach converges to the equilibrium. As such, it is important to consider the learning of the other agents as highlighted by this example.

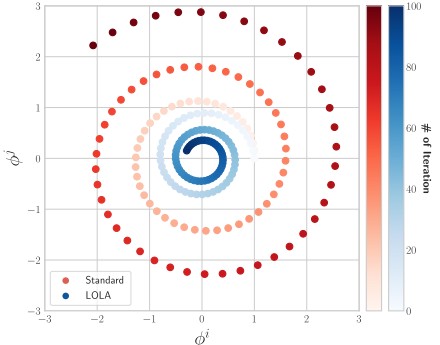

Figure 8: Learning paths on the zero-sum game. The standard approach with the stationary assumption diverges, resulting in worse performance for both agents. In contrast, an approach that considers the learning process of the other agents, such as LOLA (Foerster et al., 2018a), converges to the equilibrium.

---

[2]Available at `https://github.com/schroederdewitt/multiagent_mujoco`

# G Analysis on Joint Policy Dynamics

## G.1 IPD

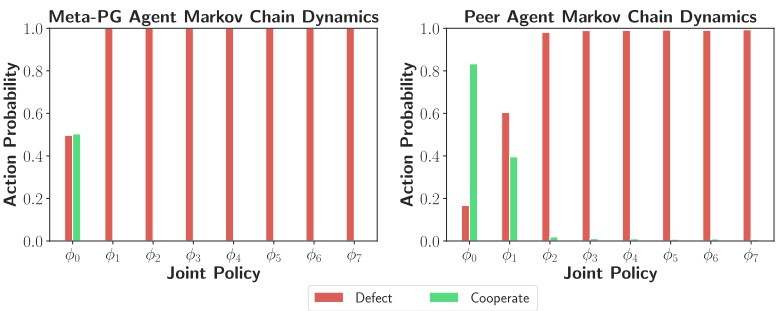

Figure 9: Action probability dynamics with Meta-PG in IPD with a cooperating persona peer

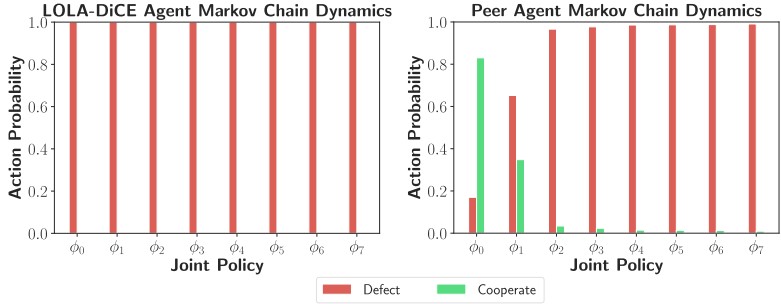

Figure 10: Action probability dynamics with LOLA-DiCE in IPD with a cooperating persona peer

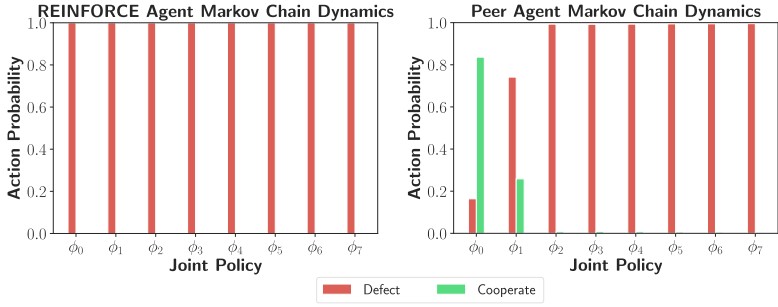

Figure 11: Action probability dynamics with REINFORCE in IPD with a cooperating persona peer

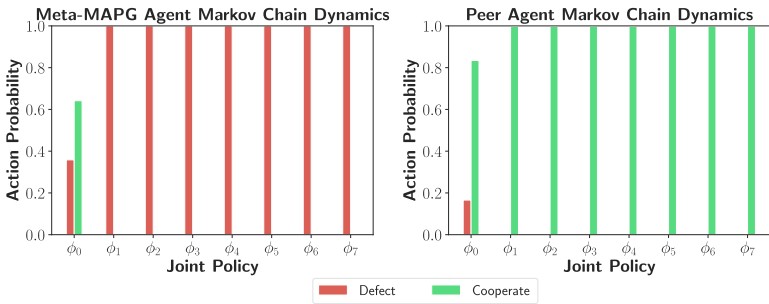

Figure 12: Action probability dynamics with Meta-MAPG in IPD with a cooperating persona peer

### G.2 RPS

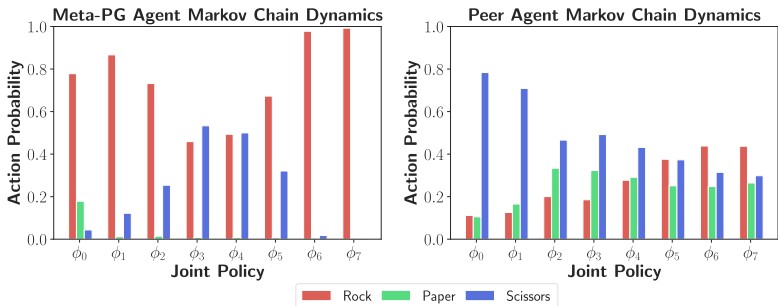

Figure 13: Action Probability Dynamics with Meta-PG in RPS with a scissors persona opponent

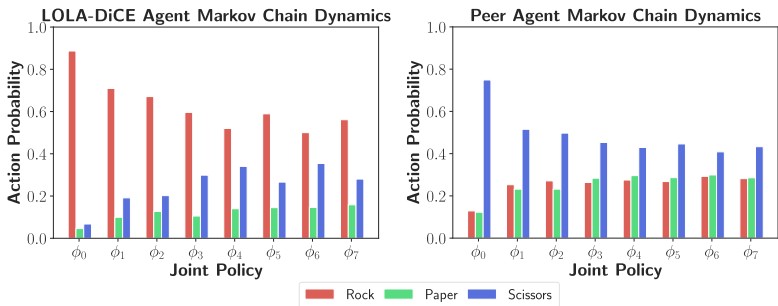

Figure 14: Action Probability Dynamics with LOLA-DiCE in RPS with a scissors persona opponent

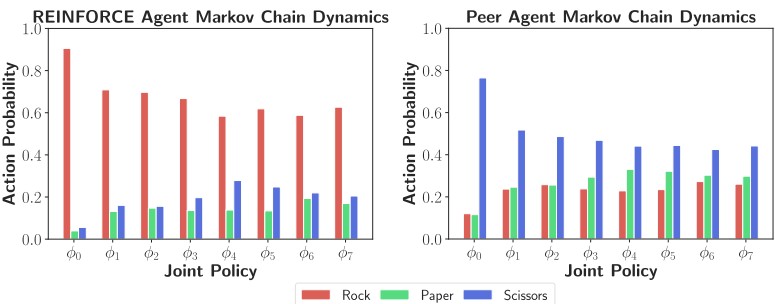

Figure 15: Action Probability Dynamics with REINFORCE in RPS with a scissors persona opponent

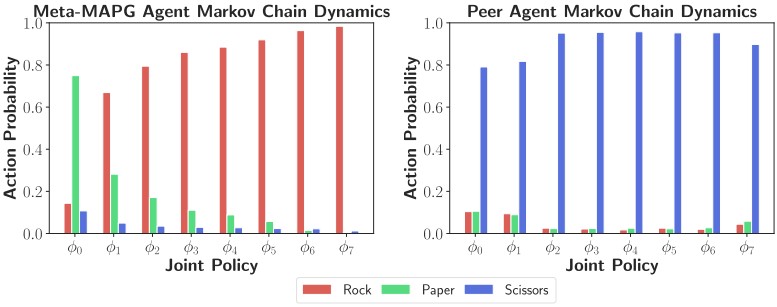

Figure 16: Action Probability Dynamics with Meta-MAPG in RPS with a scissors persona opponent

# H  HYPERPARAMETER DETAILS

We report our hyperparameter values that we used for each of the methods in our experiments:

## H.1  META-MAPG AND META-PG

| Hyperparameter | Value |
|---|---|
| Trajectory batch size $K$ | 4, 8, 16, 32, 64 |
| Number of parallel threads | 5 |
| Actor learning rate (inner) | 1.0, 0.1 |
| Actor learning rate (outer) | 1e-4 |
| Critic learning rate (outer) | 1.5e-4 |
| Episode horizon $H$ | 150 |
| Max chain length $L$ | 7 |
| GAE $\lambda$ | 0.95 |
| Discount factor $\gamma$ | 0.96 |

Table 3: IPD

| Hyperparameter | Value |
|---|---|
| Trajectory batch size $K$ | 64 |
| Number of parallel threads | 5 |
| Actor learning rate (inner) | 0.1, 0.01 |
| Actor learning rate (outer) | 1e-5 |
| Critic learning rate (outer) | 1.5e-5 |
| Episode horizon $H$ | 50 |
| Max chain length $L$ | 7 |
| GAE $\lambda$ | 0.95 |
| Discount factor $\gamma$ | 0.90 |

Table 4: RPS

| Hyperparameter | Value |
|---|---|
| Trajectory batch size $K$ | 64 |
| Number of parallel threads | 5 |
| Actor learning rate (inner) | 5e-3 |
| Actor learning rate (outer) | 5e-5 |
| Critic learning rate (outer) | 5.5e-5 |
| Episode horizon $H$ | 200 |
| Max chain length $L$ | 2 |
| GAE $\lambda$ | 0.95 |
| Discount factor $\gamma$ | 0.99 |

Table 5: 2-Agent HalfCheetah

## H.2 LOLA-DICE

| Hyperparameter | Value |
|---|---|
| Trajectory batch size $K$ | 4, 8, 16, 32, 64 |
| Actor learning rate | 1.0, 0.1 |
| Critic learning rate | 1.5e-3 |
| Episode horizon $H$ | 150 |
| Max chain length $L$ | 7 |
| Number of Look-Ahead | 1, 3, 5 |
| Discount factor $\gamma$ | 0.96 |

Table 6: IPD

| Hyperparameter | Value |
|---|---|
| Trajectory batch size $K$ | 64 |
| Actor learning rate | 0.1, 0.01 |
| Critic learning rate | 1.5e-3 |
| Episode horizon $H$ | 50 |
| Max chain length $L$ | 7 |
| Number of Look-Ahead | 1 |
| Discount factor $\gamma$ | 0.90 |

Table 7: RPS

| Hyperparameter | Value |
|---|---|
| Trajectory batch size $K$ | 64 |
| Actor learning rate | 5e-3 |
| Critic learning rate | 1.5e-4 |
| Episode horizon $H$ | 200 |
| Max chain length $L$ | 2 |
| Number of Look-Ahead | 1 |
| Discount factor $\gamma$ | 0.99 |

Table 8: 2-Agent HalfCheetah

## H.3 REINFORCE

| Hyperparameter | Value |
|---|---|
| Trajectory batch size $K$ | 4, 8, 16, 32, 64 |
| Actor learning rate | 1.0, 0.1 |
| Episode horizon $H$ | 150 |
| Max chain length $L$ | 5 |
| Discount factor $\gamma$ | 0.96 |

Table 9: IPD

| Hyperparameter | Value |
|---|---|
| Trajectory batch size $K$ | 64 |
| Actor learning rat | 0.1, 0.01 |
| Episode horizon $H$ | 50 |
| Max chain length $L$ | 7 |
| Discount factor $\gamma$ | 0.90 |

Table 10: RPS

| Hyperparameter | Value |
|---|---|
| Trajectory batch size $K$ | 64 |
| Actor learning rate | 5e-3 |
| Episode horizon $H$ | 200 |
| Max chain length $L$ | 2 |
| Discount factor $\gamma$ | 0.99 |

Table 11: 2-Agent HalfCheetah

