# OpenReview forum: "A Policy Gradient Algorithm for Learning to Learn in Multiagent Reinforcement Learning"
_ICLR.cc/2021/Conference — Reject_

### Official Review · AnonReviewer2 · 2020-10-19
**This paper addresses the non-stationary policies of other agents in MARL, by the idea of meta RL meta-trained on a distribution of other agents' policies.**

**Rating:** 5
**Confidence:** 4

**Review:**

This paper points out that a key challenge in MARL is the non-stationarity of other agents' policies, as opposed to previous papers which only account for non-stationarity of the environment. The paper extends (Al-Shedivat et al., 2018) by directly conditioning the meta-policy on a distribution of other agents' policies. In my opinion, the major contribution of this paper is a new multiagent meta learning theoretic framework that explicitly accounts  for the dynamics of all agents.

Strengths of the paper:
1) A new perspective in MARL that considers nonstationarity of MARL in terms of dynamics of the other agents' policies
2) A new theoretically grounded algorithm that explicitly models the policy dynamics of all agents

Weaknesses of the paper:
1) Except for the new perspective of incorporating policy dynamics of other agents, the backbone of the paper (i.e., meta-RL based framework to mitigate non-stationarity of MARL) is inherently the same as (Al-Shedivat et al., 2018). The novelty is somewhat limited.
2) In experiments the paper answers several questions that show the effectiveness of the new algorithm. However, this is subject to the two-agent setting. It is questionable whether such a framework can perform well in settings where there are multiple agents.

Question: does the proposed framework generalize to >2 agents scenarios? if yes, what is the reason that the authors did not conduct empirical evaluations in these scenarios?

---

> ### Author Response · Authors · 2020-11-18
> **Response to AnonReviewer2**
>
> We would like to thank the reviewer for very valuable comments. We have attempted to address individual concerns below and have carefully updated the main paper and appendix based on your feedback.
>
> > "... the backbone of the paper ... is inherently the same as (Al-Shedivat et al., 2018). The novelty is somewhat limited."
>
> Thank you for raising this issue. We would like to emphasize that our framework may at first appear to be similar to the framework by Al-Shedivat et al. (2018) because we made significant efforts to re-define their work in our general multiagent settings. Specifically, our paper makes the following theoretical contributions to build on Al-Shedivat et al. (2018):
>
> 1. **Underlying problem statement:**
> Al-Shedivat et al. (2018) bases their problem formulation off that of multi-task / continual single-agent RL. In contrast, ours is based on a general stochastic game between $n$ agents (Shapley, 1953).
> 2. **A Markov chain of joint policies:** Al-Shedivat et al. (2018) treats an evolving peer agent as an external factor, resulting in the absence of the sequential dependencies between a meta-agent's current policy and the peer agents' future policies in the Markov chain. However, our important insight is that the sequential dependencies exist in general multiagent settings as the peer agents are also learning agents based on trajectories by interacting with a meta-agent (please refer to Figure 1b).
> 3. **Meta-objective:** The meta-objective defined in Al-Shedivat et al. (2018) is based on single-agent settings (please refer to Equations 5-7 in Al-Shedivat et al. (2018)). In contrast, our meta-objective is based on general multiagent settings (please refer to Equations 2-4 in our paper).
> 4. **Meta-optimization gradient:**
> Compared to Al-Shedivat et al. (2018), our meta-optimization gradient inherently includes the additional term of the peer learning gradient that considers how an agent can directly influence the learning process of other agents.
> 5. **Importance sampling:**
> Al-Shedivat et al. (2018) requires interacting with the same peers multiple times, which is often impossible during meta-testing. Hence, the framework leverages an importance sampling correction during meta-testing. However, because the importance sampling correction generally suffers from high variance (Wang et al., 2016b), we effectively avoid using the correction by initializing from $\phi^{i}_{0}$ only once at the beginning of Markov chains for both meta-training and meta-testing.
>
> The above theoretical differences have resulted in an improved meta-agent that can learn to additionally affect future policies of other peer agents, achieving better results than the Meta-PG baseline in our experiments. We have also improved the Meta-PG baseline itself beyond its implementation in the original work to further isolate the importance of the peer learning gradient term.
> In the revision, we have worked to make the above theoretical differences clearer (please refer to Appendix D.1).
>
> > "... does the proposed framework generalize to >2 agents scenarios?"
>
> Thank you for your great suggestion about the > 2 agents scenarios. Although we should note that our $2$ agent setting matches the standard set by previous relevant work (e.g., Al-Shedivat et al. (2018), Foerster et al. (2018a)), we have conducted experiments for $n>2$ agent cases following your helpful feedback. In the revision, we have added new experiments with $3$-player and $4$-player RPS. Figure 4b shows a comparison to the Meta-PG baseline in these settings. We generally observe that the peer agents change their policies toward the mixed Nash equilibrium more quickly as the number of agents in the domain increases, resulting in decreased performance for all methods. Nevertheless, Meta-MAPG still achieves the best performance in all cases and can clearly be easily extended to settings with a greater number of agents.

---

### Official Review · AnonReviewer3 · 2020-10-27
**Review 3**

**Rating:** 6
**Confidence:** 4

**Review:**

This paper studies meta-learning in multi-agent reinforcement learning. It proposes a meta multi-agent policy gradient method that considers the learning processes of other agents in the environment for fast adaptation. This method can be seen as a unified framework of previous methods (Al-Shedivat et al. (2018) and Foerster et al. (2018a)). The method outperforms previous methods in two matrix games and 2-agent HalfCheetah.

Pros:
- The method is simple and well motivated. It additionally takes into consideration peer learning, comparing to Al-Shedivat et al. (2018).
- The method unifies the benefit of Al-Shedivat et al. (2018) and Foerster et al. (2018a).
- The method greatly outperforms these two methods in two matrix games.

Cons:
- Like LOLA, the method needs the access to policy parameters of other agents, while Al-Shedivat et al. (2018) do not. This may be impossible in mixed and competitive environments. How to deal with this?
- In experiments, most questions are answered by the two matrix games. It is not fully convinced since the state space is very limited. Why not choose RoboSumo in Al-Shedivat et al. (2018) as an experiment?
- For two matrix games, opponent policy is limited compared to complex environments, for example, halfcheetah. Although the out of distribution is tested, it is less informative for generalization. Why not test the out of distribution for halfcheetah?
- "The out of distribution has a smaller overlap between meta-train/val and meta-testing distribution." What exactly is the out of distribution?
- The experimental results need to be more elaborated. Why do Meta-PG and LOLA perform similarly to REINFORCE?

---
**After rebuttal**

The responses address my main concerns. I have increased the score to 6. But, I also agree with other reviewers that the novelty of this paper is somewhat limited.

---

> ### Author Response · Authors · 2020-11-18
> **Response to AnonReviewer3 (Part 1/2)**
>
> Thank you for your detailed review and comments about our work. We have tried to answer your individual concerns below and carefully incorporate the feedback into our revised paper.
>
> > "Like LOLA, the method needs the access to policy parameters of other agents, while Al-Shedivat et al. (2018) do not. This may be impossible in mixed and competitive environments. How to deal with this?"
>
> Thank you for your helpful feedback. Our original implementation of Meta-MAPG is centralized during meta-training but purely decentralized like Al-Shedivat et al. (2018) during meta-testing. Following your comment, we have added a decentralized meta-training algorithm with opponent modeling in the revision for settings where a meta-agent cannot access the parameters of other agents during meta-training.  Specifically, Meta-MAPG with opponent modeling computes the peer learning gradient in a decentralized manner by inferring the policy parameters of peer agents. Figure 4a of the revision shows the new result. In Question 4 of the revision, we discuss that Meta-MAPG with opponent modeling can successfully infer the policy parameters of peer agents well enough so to still significantly outperforms the Meta-PG baseline.
>
> > "... It is not fully convinced since the state space is very limited. Why not choose RoboSumo in Al-Shedivat et al. (2018) as an experiment?"
>
> Thank you for raising this important issue. We also thought it would be quite interesting to test our approach on [the RoboSumo domain](https://github.com/openai/robosumo) from Al-Shedivat et al. (2018). However, we experienced difficulties reproducing results in this domain, possibly due to the shaped reward function. Specifically, RoboSumo's shaped reward function includes a reward function for moving toward the opponent, pushing the opponent out of the center, and control. We experienced that the training performance was largely sensitive to the relative weights between these reward functions and could not reproduce the results from Al-Shedivat et al. (2018) with the default weights. Unfortunately, we believe that this reproducibility issue with the provided repository may also be a leading reason why there have not been many results on this domain in multiagent reinforcement learning research since it was originally proposed despite providing such a compelling use case.
>
> As a result, in the submission, we experimented with another challenging environment leveraging the 2-Agent HalfCheetah domain to address potential concerns about domain complexity. The 2-Agent HalfCheetah domain also has high complexity because: 1) the MuJoCo simulation has continuous and large observation/action spaces, and 2) the agents are coupled within the same robot, resulting in a more challenging control problem than controlling the robot alone with full autonomy.
>
> We have also added new experiments in the revision with a larger number of peers in RPS (please refer to Question 6 in Section 5). We note that the domain becomes harder as the number of agents in the environment increases. Specifically, the state space dimension grows exponentially as a function of the number of agents: it has a dimension of $10$ for $2$-player RPS, a dimension of $28$ for $3$-player RPS, and a dimension of $82$ for $4$-player RPS (i.e., $3^n+1$ for $n$-player RPS). Additionally, in this setting from meta-agent $i$'s perspective, $i$ needs to consider more peers in the process of affecting their future policies.
> Despite the increased difficulty, we show that Meta-MAPG can scale to this domain and still outperform the Meta-PG baseline.

---

> > ### Author Response · Authors · 2020-11-18
> > **Response to AnonReviewer3 (Part 2/2)**
> >
> > > "Why not test the out of distribution for halfcheetah?"
> >
> > Thank you for pointing out that the 2-Agent HalfCheetah experimental settings may be confusing. The reported HalfCheetah results in Figure 2d are actually based on an out of distribution evaluation setting. Specifically, Figure 7 in the appendix shows the distribution of the teammate $j$'s initial expertise. We note that we constructed the distribution with a significant gap to ensure that the meta-testing distribution is sufficiently different from the meta-training/validation distribution. This allows us to effectively evaluate the ability of our approach to generalize its learning. We see now that the fact that we did not provide an analogous in distribution testing case may have led to confusion about this experiment. We will add an in distribution result to the final draft as well to promote increased clarity.
> >
> > > "What exactly is the out of distribution?"
> >
> > Thank you for your question about the out of distribution. In the paper, we have defined the out of distribution when there is a small overlap between the peers' policies used for meta-training/validation and those used for meta-testing. For example, we refer to the case in Figure 3c as the out of distribution because the meta-testing personas have a small overlap with the meta-training/validation personas. Following your helpful feedback, we have updated Section 5 and made this more clear.
> >
> > > "Why do Meta-PG and LOLA perform similarly to REINFORCE?"
> >
> > Thank you for your helpful suggestion regarding the baselines. To explore each adaptation method in more detail, in the revision, we added visualizations of the action probability dynamics throughout the Markov chain (please refer to Figures 9-12 for IPD and Figures 13-16 for RPS, and Questions 1 and 5 for discussion). In general, we observe that the baseline methods (Meta-PG, LOLA-DiCE, REINFORCE) have effectively converged to win against the other agent $j$ in the first few trajectories. For instance, an agent $i$ has a high probability of choosing rock when playing against $j$ with initially a high probability of choosing scissors in RPS.
> > This strategy, however, results in the opponent quickly changing its behavior toward the mixed Nash equilibrium strategy of $(1/3, 1/3, 1/3)$ for the rock, paper, and scissors probabilities.

---

### Official Review · AnonReviewer4 · 2020-10-30
**Unclear experimental setup**

**Rating:** 6
**Confidence:** 5

**Review:**

This paper makes a full derivation of the meta-learning gradient estimation for multi-agent adaptation.
The resulting algorithm combines the meta-learning of the opponent's updates (existing in LOLA) and of oneself's futur updates (existing in Meta-PG).

While the theoretical part of the paper is clear and well explained, the experimental setup is missing a lot of details to be interpreted:
- In each experiment, it seems (but never explicitly formulated) that "agent i" (agent 1, since all experiments are involving 2 players) is doing the meta-learning algorithm (meta-MAPG, meta-PG or LOLA) while the other (agent 2) is a naive agent initialised with defective/cooperative policies.
- In that case: how are naive agent updated? with simple policy-gradient?
- How many lookahead are used (denoted by L in algorithms)?
- Why did LOLA failed at learning to cooperate with the cooperative opponents? (it should have learned to cooperate, unless naive agents are still doing selfish PG updates --and in that case, meta-MAPG results are very impressive)?
- Are the opponent's policies given or learned (i.e. with opponent modelling)?

Also, I would have been interesting to see an ablation study showing the importance of the "own learning" and "peer learning" terms in equation 6 (from the same implementation with fixed HP). Does the authors have tried it?

---

> ### Author Response · Authors · 2020-11-18
> **Response to AnonReviewer4**
>
> We would like to thank the reviewer for very insightful comments. We have attempted to address individual concerns below and have updated the main paper and appendix accordingly.
>
> > "... agent $i$ ... is doing the meta-learning algorithm ... while the other (agent 2) is a naive agent ..."
>
> > "In that case: how are naive agent updated? with simple policy-gradient?"
>
> Thank you for pointing out your confusion about the experimental setup. Each method (either Meta-MAPG, Meta-PG, LOLA-DiCE, or REINFORCE) is evaluated by allowing it to update the policy of agent $i$. Meanwhile, the peer agent $j$ is initialized with some persona (e.g., the cooperative persona in IPD) and updates its policy leveraging a learning process based on the policy gradient with a linear feature baseline (Duan et al., 2016a). We have updated the main text in Section 5 to clarify our experimental setup.
>
> > "How many lookahead are used (denoted by L in algorithms)?"
>
> Thank you for your comment. We have used the max chain length $L$ of 7 for IPD and RPS, and we used $L$ of 2 for 2-agent HalfCheetah (please refer to Appendix H for remaining hyperparameter details). Following your helpful feedback, we have updated Section 5 and made this more clear.
>
> > "Why did LOLA failed at learning to cooperate with the cooperative opponents?"
>
> Thank you for your interesting question regarding the LOLA baseline. It is correct that if there are two LOLA agents, then the agents have been shown to learn to cooperate in IPD. However, the peer agent $j$ in our setting updates its policy based on the (naive) policy gradient with a linear baseline. Hence, our experimental setup is closer to the LOLA-Ex vs. Naive Learning (NL-Ex) experiment in Table 4 of Foerster et al. (2018a), where the LOLA agent gains larger rewards than the other naive learning agent by choosing to defect with higher probability.
>
> We have observed similar behavior for the LOLA method in our experimental results. Specifically, after meta-training, the LOLA agent has effectively converged to act with a high probability of selecting the defect action, attempting to get larger rewards than the peer agent $j$ (regardless of whether $j$ is initialized with the cooperative or defecting persona). While this strategy can result in better initial performance than the peer, $j$ will quickly change its policy so that it defects with a high probability. To show how this happens explicitly, we have added visualizations about the action probability dynamics throughout the Markov chain in the revision (please refer to Figures 9-12 for IPD and Figures 13-16 for RPS, and Questions 1 and 5 for discussion).
>
> > "Are the opponent's policies given or learned (i.e. with opponent modelling)?"
>
> Thank you for your helpful feedback. Our original implementation of Meta-MAPG was centralized (i.e., requiring the true policy parameters of other agents) during meta-training but purely decentralized like Al-Shedivat et al. (2018) during meta-testing (please refer to the algorithm in Section 3.3 for more details). However, there are certainly settings in which a meta-agent cannot access the policy parameters of other agents during meta-training. Hence, following your comment, we have added a decentralized meta-training algorithm with opponent modeling that computes the peer learning gradient while leveraging only approximated peer policy parameters. Figure 4a of the revision shows the new result. In Question 4 of the revision, we discuss that Meta-MAPG with opponent modeling can successfully infer the policy parameters of peer agents well enough so to still significantly outperforms the Meta-PG baseline.
>
> > "... an ablation study showing the importance of the "own learning" and "peer learning" terms in equation 6 ..."
>
> Following your suggestion, we have added an ablation study in the revision that compares Meta-MAPG against two methods: one method trained without the peer learning gradient and the other trained without the own learning gradient. The result in Figure 4c shows that training without the peer learning term results in a meta-agent that cannot exploit the learning process of peer agents. Also, a meta-agent trained without the own learning term cannot effectively change its learning over time to quickly adapt to changing peers. By contrast, Meta-MAPG achieves far better performance than both ablated baselines by effectively accounting for both its own learning process and the learning process of other agents in the environment.

---

### Official Review · AnonReviewer1 · 2020-11-02
**solid paper, evaluation could be more ambitious**

**Rating:** 6
**Confidence:** 3

**Review:**

This paper considers the problem of meta-learning in a multi-agent environment under the assumptions that:
* the learning agent's policy evolves over time as a function of the other agent's actions
* the other agents' policies evolve potentially using the learning agent's actions.
The policy learning problem is assumed to be Markovian. The meta-learning problem is considered to be that
of finding the best initial policy parameters (that will subsequently be evolved according to the learning dynamics) as to maximize the agent's cumulative marginal payoff.

The paper is very well written, easy to read and relatively straightforward in its exposition. I do not have any big remarks about writing except that the authors may want to rethink the term "defective persona" to avoid the weird double meaning. A sufficient amount of related work presented and the lineage of the ideas is traced convincingly well.

The main contribution of this paper is to extend the ideas of Al-Shedivat et al. in a way that exposes the other agent's learning dynamics to the policy optimization (as opposed to treating them as a non-stationarity). The policy gradient form corresponding to this setting is derived in Theorem 1. The approach is evaluated in a synthetic experiment using iterated games as well as a somewhat less synthetic experiment on a quarter-cheetah problem (each agent controls a leg of the half-cheetah).

I think that while the paper is incremental, the point that is raised within is rather intriguing. If anything my main criticism is that the authors could have gone for a more challenging setting that iterated games. E.g. recent results (https://arxiv.org/pdf/1901.08654.pdf) indicate that in settings like collaborative exploration, being aware of the other player's learning dynamics is important for achieving a better outcome.  Perhaps the policy gradient approach can solve issues that cannot be addressed straightforwardly within the bandit framework. Another question is whether the approach can be used successfully to tune the inner learning process, e.g. by incorporating the policy gradient step size and other hyper-parameters into phi_0.

Overall I think this is a solid paper, which would benefit significantly from more ambitious problems.

---

> ### Author Response · Authors · 2020-11-18
> **Response to AnonReviewer1**
>
> Thank you for your helpful review and detailed questions. We have attempted to address each comment individually below and have carefully incorporated this feedback into an update to our main paper and appendix.
>
> > "... the authors may want to rethink the term "defective persona" to avoid the weird double meaning."
>
> Thank you for pointing out that the terminology used in our submitted draft may be confusing. To avoid the double meaning, we have clarified the term in the revision by changing each instance of ["defective"](https://dictionary.cambridge.org/us/dictionary/english/defective) to ["defecting"](https://dictionary.cambridge.org/us/dictionary/english/defecting) (e.g., "defective persona" to "defecting persona").
>
> > "...  a more challenging setting that iterated games."
>
> Thank you for raising this important issue. To address potential concerns about domain complexity, we experimented with a challenging environment leveraging the 2-Agent HalfCheetah domain in the submission. The 2-Agent HalfCheetah domain has high complexity because: 1) the MuJoCo simulation has continuous and large observation/action spaces, and 2) the agents are coupled within the same robot, resulting in a more challenging control problem than controlling the robot alone with full autonomy.
>
> We have also added new experiments in the revision with a larger number of peers in RPS (please refer to Question 6 in Section 5). We note that the domain becomes harder as the number of agents in the environment increases. Specifically, the state space dimension grows exponentially as a function of the number of agents: it has a dimension of $10$ for $2$-player RPS, a dimension of $28$ for $3$-player RPS, and a dimension of $82$ for $4$-player RPS (i.e., $3^{n}+1$ for $n$-player RPS). Additionally, in this setting from a meta-agent $i$'s perspective, $i$ needs to consider more peers in shaping their future policies. Despite the increased difficulty, we show that Meta-MAPG can scale to this domain and still outperform the Meta-PG baseline.
>
> Our future work includes extending our approach to real-world scenarios, such as [the suggested collaborative exploration between multiple agents](https://arxiv.org/pdf/1901.08654.pdf). We have updated Section 6 in the revision accordingly.
>
> > "Another question is whether the approach can be used successfully to tune the inner learning process, e.g. by incorporating the policy gradient step size ..."
>
> Thank you for your question. To clarify, as suggested in Al-Shevidat et al. (2018), we also have learned dynamic inner-loop learning rates during meta-training to tune the inner learning process. We mentioned this detail in the appendix of the submitted draft. However, we clarified this point in the main text based on your feedback (please refer to Section 5 in the revision).

---

### Author Response · Authors · 2020-11-18
**Response to all (Revision 1)**

We would like to thank all of the reviewers for their valuable time and helpful comments. In Revision 1, we have carefully considered the feedback and made adjustments to the main paper and appendix accordingly. Here, we summarize new results in Revision 1 with individual comments for each reviewer to follow.

**Summary of new results:**
1. **Opponent modeling:**
Our original implementation of Meta-MAPG in the submitted paper was centralized during meta-training and decentralized during meta-testing. In the revision, we additionally provide a decentralized meta-training algorithm with opponent modeling that computes the peer learning gradient using only an approximation of the policy parameters for peer agents. The new result with opponent modeling is shown in Figure 4a. We discuss the result in Question 4 and explain the approach in detail in Appendix B.
2. **Analysis of Strategies Learned by Baselines and Meta-MAPG:**
We examine each adaptation method in more detail by visualizing the action probability dynamics throughout the Markov chain in IPD (Figures 9-12) and RPS (Figures 13-16). We discuss these results in Questions 1 and 5 of the updated main paper.
3. **Larger number of peers:**
We experiment with $3$-player and $4$-player RPS with $2$ and $3$ peers, respectively, in the revision. The new empirical results with a larger number of actively learning peer agents are shown in Figure 4b, and we explain these findings in Question 6.
4. **Ablation study:**
We conduct an ablation study and compare Meta-MAPG against two methods: one method trained while excluding the own learning gradient and another agent trained while excluding the peer learning gradient.
We detail the results of this ablation study in Figure 4c and Question 7.

---

### Author Response · Authors · 2020-11-23
**Following up for feedback on our revision**

We really appreciate all your very helpful reviews of our submitted paper and would love to receive feedback after the substantial revisions we have made. Specifically, we have added new experiments, analyses, and clarifications in the revision to address the initial concerns mentioned by the reviewers. Please let us know if you may have any additional follow-up questions. We are happy to address them within the rebuttal period and hope that we can rectify any issues that may cause the reviewers to feel like our paper should not be accepted. Thank you all for your valuable time and kind consideration.

---

### Decision · Program_Chairs · 2021-01-07
**Final Decision**

**Decision:**

Reject

**Comment:**

This paper studies the problem of multi-agent meta-learning. It can be viewed as extending Al-Shedivat et al. (2018) by incorporating the dynamics of other agents. The reviewers praised clear writing and theory. There were two main concerns. The first concern is the novelty when compared to Al-Shedivat et al. (2018). The second concern are experiments, which could be more ambitious and are not always clearly described.

The reviews of this paper were borderline and this was not enough to get accepted.